# Optimal transport for automatic alignment of untargeted metabolomic data

Marie Breeur[1†], George Stepaniants[2†], Pekka Keski-Rahkonen[1], Philippe Rigollet[2], Vivian Viallon[1]*

[1]Nutrition and Metabolism Branch, International Agency for Research on Cancer, Lyon, France; [2]Massachusetts Institute of Technology, Department of Mathematics, Boston, United States

**\*For correspondence:**
viallonv@iarc.who.int

[†]These authors contributed equally to this work

**Competing interest:** The authors declare that no competing interests exist.

**Abstract** Untargeted metabolomic profiling through liquid chromatography-mass spectrometry (LC-MS) measures a vast array of metabolites within biospecimens, advancing drug development, disease diagnosis, and risk prediction. However, the low throughput of LC-MS poses a major challenge for biomarker discovery, annotation, and experimental comparison, necessitating the merging of multiple datasets. Current data pooling methods encounter practical limitations due to their vulnerability to data variations and hyperparameter dependence. Here, we introduce Gromov-Matcher, a flexible and user-friendly algorithm that automatically combines LC-MS datasets using optimal transport. By capitalizing on feature intensity correlation structures, GromovMatcher delivers superior alignment accuracy and robustness compared to existing approaches. This algorithm scales to thousands of features requiring minimal hyperparameter tuning. Manually curated datasets for validating alignment algorithms are limited in the field of untargeted metabolomics, and hence we develop a dataset split procedure to generate pairs of validation datasets to test the alignments produced by GromovMatcher and other methods. Applying our method to experimental patient studies of liver and pancreatic cancer, we discover shared metabolic features related to patient alcohol intake, demonstrating how GromovMatcher facilitates the search for biomarkers associated with lifestyle risk factors linked to several cancer types.

## eLife assessment

The authors describe an **important** tool, GromovMatcher, that can be used to compare proteomic data from various experimental approaches. The underlying method is innovative, the algorithm is clearly described, and the validation that is presented is **convincing**.

## Introduction

Untargeted metabolomics is a powerful analytical technique used to identify and measure a large number of metabolites in a biological sample without preselecting targets (*Patti, 2011*). This approach allows for a comprehensive overview of an individual's metabolic profile, provides insights into the biochemical processes involved in cellular and organismal physiology (*Wishart, 2019*; *Pirhaji et al., 2016*), and allows for the exploration of how environmental factors impact metabolism (*Rappaport et al., 2014*; *Bedia, 2022*). It creates new opportunities to investigate health-related conditions, including diabetes (*Wang et al., 2011*), inflammatory bowel diseases *Franzosa et al., 2019*, and various cancer types (*Loftfield et al., 2021*; *Li et al., 2020*). However, a major challenge in biomarker discovery, metabolic signature identification and other untargeted metabolomic analyses lies in the

low throughput of experimental data, necessitating the development of efficient pooling algorithms capable of merging datasets from multiple sources (*Loftfield et al., 2021*).

A common experimental technique in untargeted metabolomics is liquid chromatography-mass spectrometry (LC-MS) which assembles a list of thousands of unlabeled metabolic features characterized by their mass-to-charge ratio (*m/z*), retention time (RT; *Zhou et al., 2012*), and intensity across all biological samples. Combining LC-MS datasets from multiple experimental studies remains challenging due to variation in the *m/z* and RT of a feature from one study to another (*Zhou et al., 2012*; *Ivanisevic and Want, 2019*). This problem is further compounded by differing instruments and analytical protocols across laboratories, resulting in seemingly incompatible metabolomic datasets.

Manual matching of metabolic features can be a laborious and error-prone task (*Loftfield et al., 2021*). To address this challenge, several automated methods have been developed for metabolic feature alignment. One such method is MetaXCMS, which matches LC-MS features based on user-defined *m/z* and RT thresholds (*Tautenhahn et al., 2011*). More advanced tools use information on feature intensities measured in samples. For instance, PAIRUP-MS uses known shared metabolic features to impute the intensities of all features from one dataset to another *Hsu et al., 2019*. Metab-Combiner (*Habra et al., 2021*) and M2S (*Climaco Pinto et al., 2022*) compare average feature intensities, along with their *m/z* and RT values, to align datasets without requiring extensive knowledge of shared features. These automated alignment methods have accelerated our ability to pool and annotate datasets as well as extract biologically meaningful biomarkers. However, they demand substantial fine-tuning of user-defined parameters and ignore correlations among metabolic features which provide a wealth of additional information on shared features.

Here, we introduce GromovMatcher, a user-friendly flexible algorithm which automates the matching of metabolic features across experiments. The main technical innovation of GromovMatcher lies in its ability to incorporate the correlation information between metabolic feature intensities, building upon the powerful mathematical framework of computational optimal transport (OT; *Peyré and Cuturi, 2019*; *Villani, 2021*). OT has proven effective in solving various matching problems and has found applications in multiomics analysis (*Demetci et al., 2022*), cell development (*Schiebinger et al., 2019*; *Yang et al., 2020*), and chromatogram alignment (*Skoraczyński et al., 2022*). Here, we leverage the Gromov-Wasserstein (GW) method (*Mémoli, 2011*; *Solomon et al., 2016*), which matches datasets based on their distance structure and has been seminally applied to spatial reconstruction problems in genomics *Nitzan et al., 2019*. GromovMatcher builds upon the GW algorithm to automatically uncover the shared correlation structure among metabolic feature intensities while also incorporating *m/z* and RT information in the final matching process.

To assess the performance of GromovMatcher, we systematically benchmark it on synthetic data with varying levels of noise, feature overlap, and data normalizations, outperforming prior state-of-the-art methods of metabCombiner (*Habra et al., 2021*) and M2S (*Climaco Pinto et al., 2022*). Next, we apply GromovMatcher to align experimental patient studies of liver and pancreatic cancer to a reference dataset and associate the shared metabolic features to each patient's alcohol intake. Through these efforts, we demonstrate how GromovMatcher data pooling improves our ability to discover biomarkers of lifestyle risk factors associated with several types of cancer.

## Results
### GromovMatcher algorithm

GromovMatcher uses the mathematical framework of OT to find all matching metabolic features between two untargeted metabolomic datasets (*Figure 1*). It accepts two LC-MS datasets with possibly different numbers of metabolic features and samples. Each feature, fx$_i$ in Dataset 1 and fy$_j$ in Dataset 2, is identified by its *m/z*, RT, and vector of feature intensities across samples (*Figure 1a*). The primary tenet of GromovMatcher is that shared metabolic features have similar correlation patterns in both datasets and can be matched based on the distance/correlations between their feature intensity vectors. Specifically, GromovMatcher computes the pairwise distances between the feature intensity vectors of each metabolic feature in a dataset and saves them into a distance matrix, one per dataset (*Figure 1b*). In practice, we use either the Euclidean distance or the cosine distance (negative of correlation) to perform this step (Materials and methods). The resulting distance matrices contain information about the feature intensity similarity within each study. Using optimal transport, we can

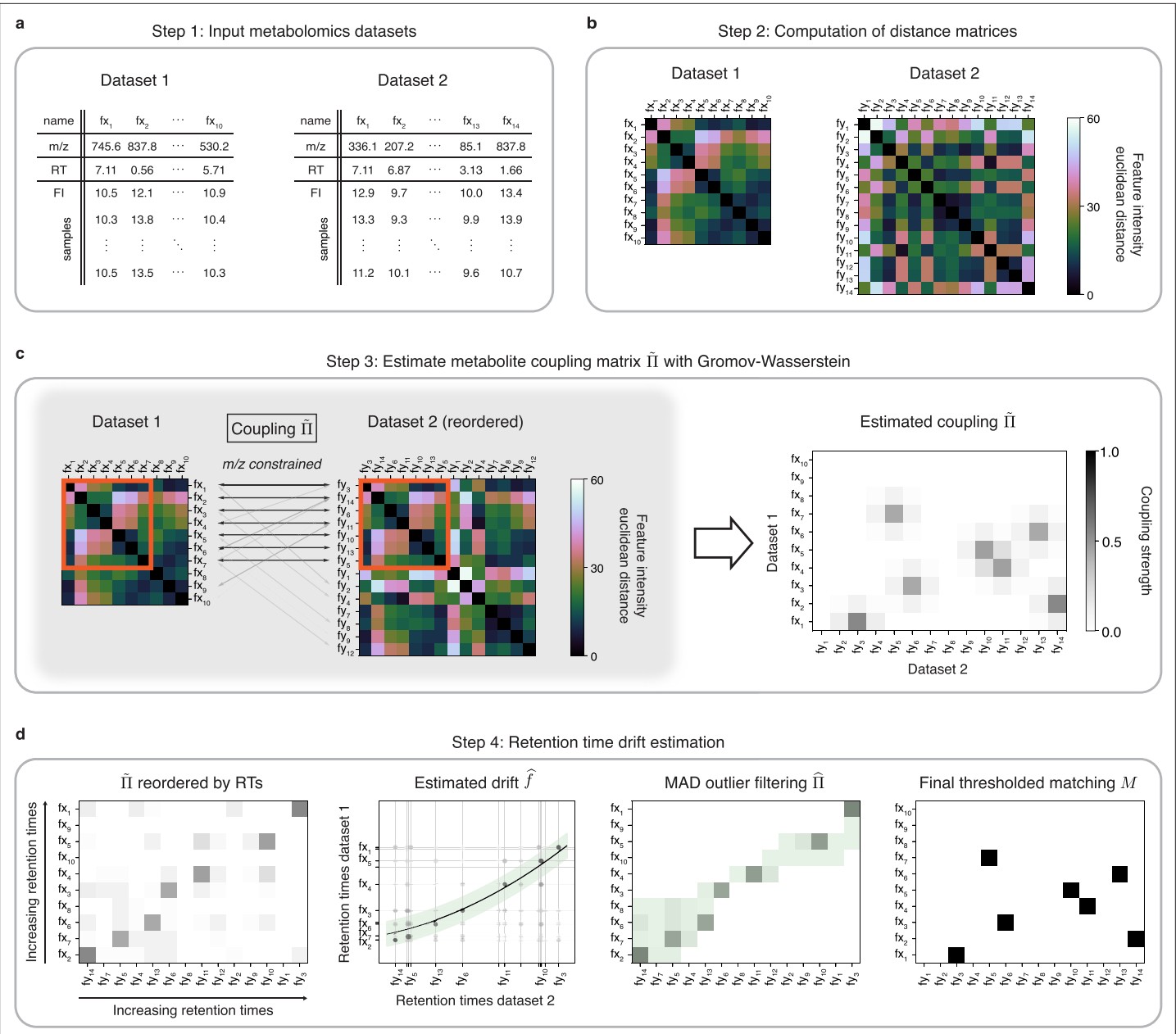

**Figure 1.** An optimal transport approach for combining untargeted metabolomics datasets (GromovMatcher). (**a**) Inputs are two LC-MS datasets of unlabeled metabolic features (rows) identified by their *m/z*, RT, and feature intensities across biospecimen samples. Both studies can have differing numbers of metabolic features and samples. (**b**) In both datasets, the intensities across samples of each metabolic feature are formed into a vector and Euclidean distances between these feature vectors are computed and stored in a distance matrix. (**c**) Based on the technique of optimal transport, the unbalanced GW algorithm learns a coupling matrix $\tilde{\Pi}$ that places large weights $\tilde{\Pi}_{ij} \geq 0$ when $\mathbf{fx}_i$ and $\mathbf{fy}_j$ likely correspond to the same metabolic feature. It optimizes $\tilde{\Pi}$ to match features with similar pairwise distances (red outlined boxes) whose *m/z* ratios are close. (**d**) The final step of GromovMatcher plots the retention times of features from both datasets against each other and fits a spline interpolation $\hat{f}$ weighted by the estimated coupling weights $\tilde{\Pi}$. This retention time drift function is then used to set all entries $\tilde{\Pi}_{ij}$ to zero for those outlier pairs $(\mathbf{fx}_i, \mathbf{fy}_j)$ which exceed twice the median absolute deviation (MAD) around $\hat{f}$ (green highlighted region). Finally, the coupling matrix $\tilde{\Pi}$ is filtered and/or thresholded to obtain a refined coupling $\widehat{\Pi}$ which is then binarized to obtain a one-to-one matching $M$ between a subset of metabolite pairs in both datasets.

deduce shared subsets of metabolic features in both datasets which have corresponding feature intensity distance structures.

OT was originally developed to optimize the transportation of soil for the construction of forts (*Monge, 1781*) and was later generalized through the language of probability theory and linear

programming (*Kantorovich, 2006*), leading to efficient numerical algorithms and direct applications to planning problems in economics. The ability of OT to efficiently match source to target locations found applications in data science for the alignment of distributions (*Courty et al., 2017*; *Alvarez-Melis et al., 2019*) and was generalized by the Gromov-Wasserstein (GW) method (*Peyré et al., 2016*; *Alvarez-Melis and Jaakkola, 2018*) to align datasets with features of differing dimensions.

In practice, a sizeable fraction of the metabolic features measured in one study may not be present in the other. Hence, in most cases only a subset of features in both datasets can be matched. Recent GW formulations for unbalanced matching problems (*Sejourne et al., 2021*) allow for matching only subsets of metabolic features with similar intensity structures (*Figure 1c*). To incorporate additional feature information, we modify the optimization objective of unbalanced GW to penalize feature matches whose $m/z$ differences exceed a fixed threshold (Materials and methods, Appendix 1). The optimization of this objective computes a *coupling matrix* $\tilde{\Pi}$ where each entry $\tilde{\Pi}_{ij} \geq 0$ indicates the level of confidence in matching metabolic feature $fx_i$ in Dataset 1 to $fy_j$ in Dataset 2.

Differences in experimental conditions can induce variations in RT between datasets that can be nonlinear and large in magnitude (*Zhou et al., 2012*; *Climaco Pinto et al., 2022*; *Habra et al., 2021*). In the spirit of previous methods for LC-MS batch or dataset alignment (*Smith et al., 2006*; *Brunius et al., 2016*; *Liu et al., 2020*; *Vaughan et al., 2012*; *Habra et al., 2021*; *Climaco Pinto et al., 2022*; *Skoraczyński et al., 2022*), the learned coupling $\tilde{\Pi}$ is used to estimate a nonlinear map (drift function) between RTs of both datasets by weighted spline regression, which allows us to filter unlikely matches from the coupling matrix to obtain a refined coupling matrix $\hat{\Pi}$ (*Figure 1d*, Materials and methods). An optional thresholding step removes matches with small weights from the coupling matrix. The final output of GromovMatcher is a binary matching matrix $M$ where $M_{ij}$ is equal to 1 if features $fx_i$ and $fy_j$ are matched and 0 otherwise. Throughout the paper, we refer to the two variants of GromovMatcher, with and without the optional thresholding step as GMT and GM respectively.

## Validation on ground-truth data

We first evaluate the performance of GromovMatcher using a real-world untargeted metabolomics study of cord blood across 499 newborns containing 4712 metabolic features characterized by their $m/z$, RT, and feature intensities (*Alfano et al., 2020*). To generate ground-truth data, we randomly divide the initial dataset into two smaller datasets sharing a subset of features (*Figure 2*). We simulate diverse acquisition conditions by adding noise to the $m/z$ and RT of dataset 2, and to the feature intensities in both datasets. Moreover, we introduce an RT drift in dataset 2 to replicate the retention time variations observed in real LC-MS experiments (Materials and methods). For comparison, we also test M2S (*Climaco Pinto et al., 2022*) and metabCombiner (*Habra et al., 2021*), both of which use $m/z$, RT, and median or mean feature intensities to match features (*Figure 3*). MetabCombiner is supplied with 100 known shared metabolic features to automatically set its hyperparameters, while M2S parameters are manually fine-tuned to optimize the F1-score in each scenario (Appendix 2). We assess the performance of GM, GMT, metabCombiner, and M2S across 20 randomly generated dataset pairs in terms of their precision (fraction of true matches among the detected matches) and recall/sensitivity (fraction of true matches detected) averaged across 20 dataset pairs.

To investigate how the number of shared features affects dataset alignment, we generate pairs of LC-MS datasets with low, medium, and high feature overlap (25%, 50%, and 75%), while maintaining a medium noise level (Materials and methods). Here, we find that GM and GMT generally outperform existing alignment methods, with a recall above 0.95 while metabCombiner and M2S tend to be less sensitive (*Figure 3b*). All methods drop in precision as the feature overlap is decreased, with GM and GMT still maintaining an average precision above 0.8.

Next we evaluate all four methods at low, moderate, and high noise levels for pairs of datasets with 50% overlap in their features (Materials and methods). Our results show that GMT, GM, and M2S maintain an average recall above 0.89, while metabCombiner's recall drops below 0.6 for high noise. At large noise levels, RT drift estimation becomes more challenging, leading to a higher rate of false matches between metabolites (lower precision) for all four methods (*Figure 3—figure supplement 1*). Nevertheless, GMT obtains a high average precision and recall of 0.86 and 0.92, respectively.

A notable difference between GM, metabCombiner, and M2S lies in their use of feature intensities. MetabCombiner expects that the mean feature intensity rankings are identical across studies, while M2S assumes that shared features have similar median intensities. In contrast, GM uses both the mean

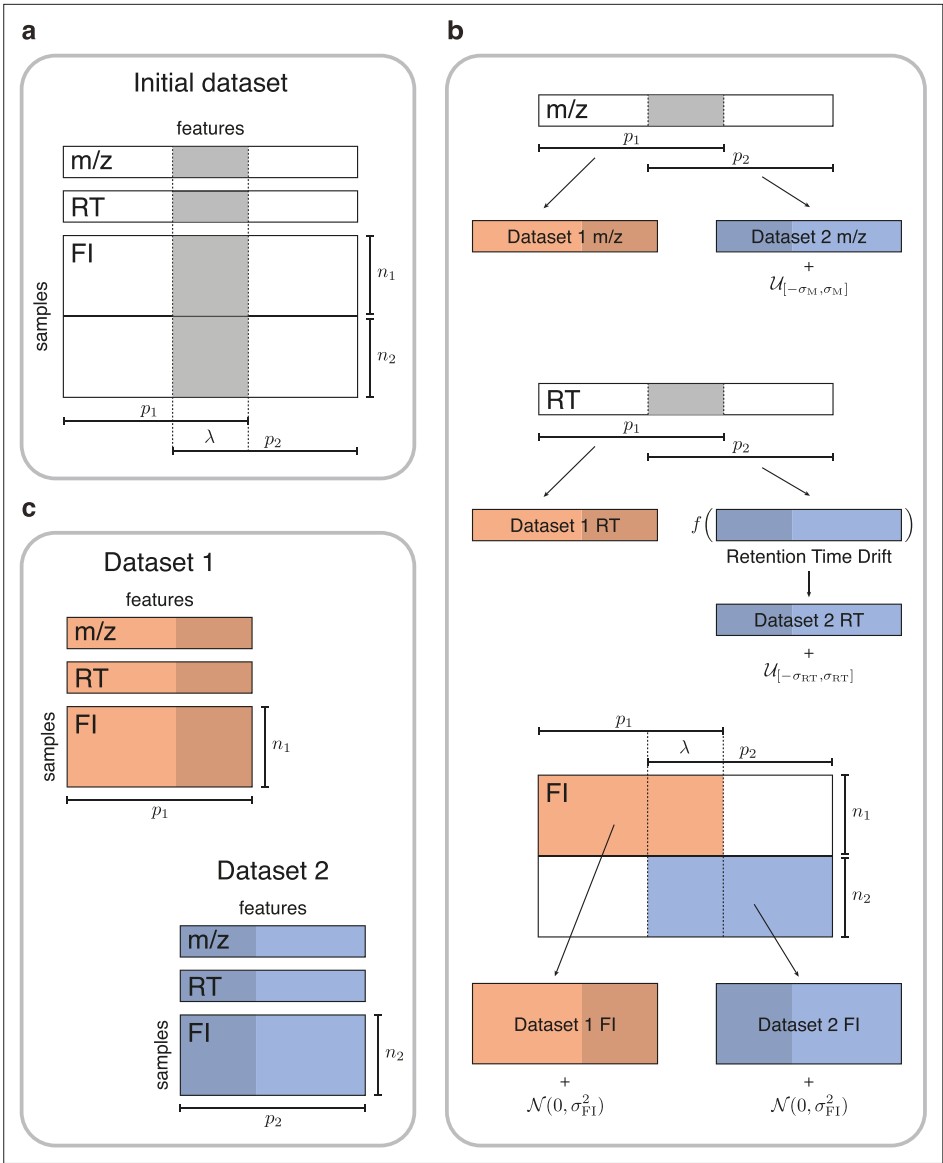

**Figure 2.** Simulated data for testing untargeted metabolomics alignment methods. (**a**) Initial LC-MS dataset taken from the EXPOsOMICS project with *m/z*, RT, and feature intensities of $p = 4,712$ metabolites identified in cord blood across $n = 499$ newborns. (**b**) Newborns (rows) are split into two disjoint groups of sizes $n_1 = 249$ and $n_2 = 250$ respectively and metabolic features (columns) are split into two equal groups of size $p_1 = p_2$ with overlap $\lambda p$ where $\lambda = 0.25, 0.5, 0.75$ (Materials and methods). Datasets are perturbed by additive noise of magnitude $(\sigma_M, \sigma_{RT}, \sigma_{FI})$ and a nonlinear drift $f(x)$ is applied to the RTs of dataset 2. (**c**) The two resulting datasets share $\lambda = 25\%, 50\%$, or $75\%$ of the original dataset's metabolic features.

feature intensities and their variances and covariances. In practice, differences in experimental assays or study populations can lead to greater variation in feature intensities, making matchings based on these statistics less reliable. Centering and scaling the feature intensities to unit variance avoids potential biases arising from inconsistent feature intensity magnitudes, but preserves correlations that GM leverages.

Exploring this further, we test how sensitive all four methods are to centering and scaling of feature intensities. MetabCombiner and M2S are tuned using the same methodology as for non-centered and non-scaled data. For M2S, we match features solely based on their *m/z* and RT. In this experiment (*Figure 3—figure supplement 2*), the absence of intensity magnitude information significantly affects metabCombiner's performance and, to a lesser extent, M2S. GM and GMT still obtain accurate matchings, due to their use of correlation structures which are preserved under centering and scaling.

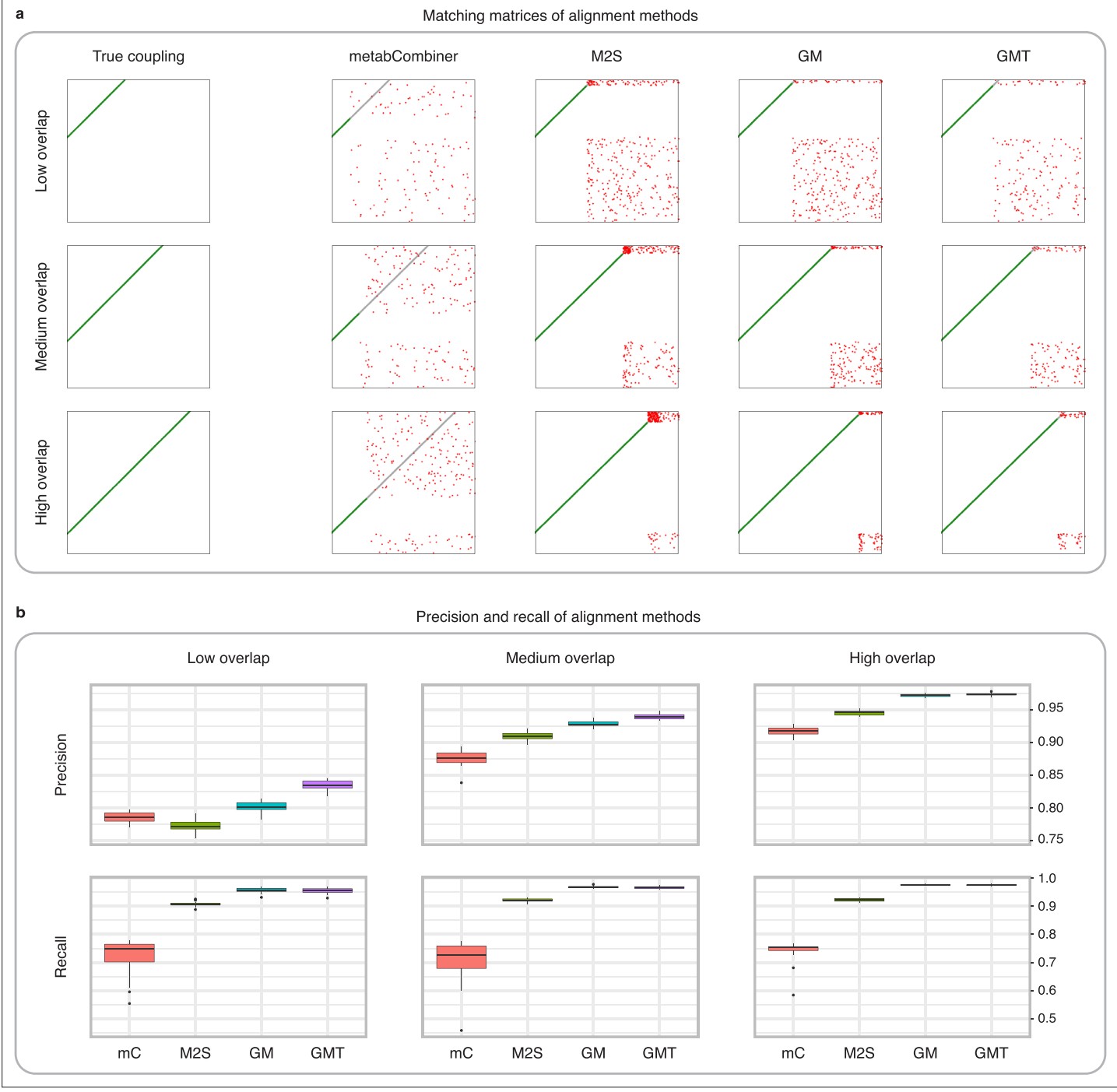

**Figure 3.** Comparison of MetabCombiner, M2S, and GromovMatcher on simulated data. (**a**) Ground-truth matchings, and matchings inferred by metabCombiner, M2S, GM, and GMT. Pairs of datasets are generated for three levels of overlap (low, medium and high), with a medium noise level (Materials and methods). Matches correctly recovered (true positives) are represented in green. True matches that are not recovered (false negatives) are highlighted in grey. Incorrect matches (false positives) are plotted in red. Features in rows and columns of matching matrices are reordered for visual clarity. (**b**) Average precision and recall on 20 randomly generated pairs of datasets, for three levels of overlap (low, medium, and high) with a medium noise level.

The online version of this article includes the following figure supplement(s) for figure 3:

**Figure supplement 1.** Average precision and recall obtained on simulated data, with fixed overlap $\lambda = 0.5$.

**Figure supplement 2.** Performance on centered and scaled data.

## Application to EPIC data

Next, we apply GM, metabCombiner and M2S to align datasets from the European Prospective Investigation into Cancer and Nutrition (EPIC) cohort, a prospective study conducted across 23 European centers. EPIC comprises more than 500,000 participants who provided blood samples at recruitment (*Riboli et al., 2002*). Untargeted metabolomics data were successively acquired in several studies nested within the full cohort.

In the present work, we use LC-MS data from the EPIC cross-sectional (CS) study (*Slimani et al., 2003*) and two matched case-control studies nested within EPIC, on hepatocellular carcinoma (HCC; *Stepien et al., 2016*; *Stepien et al., 2021*) and pancreatic cancer (PC; *Gasull et al., 2019*). LC-MS untargeted metabolomic data were acquired at the International Agency for Research on Cancer, making use of the same platform and methodology (Materials and methods). The number of samples and features in each study is displayed in *Figure 4a*.

*Loftfield et al., 2021* previously matched features from the CS, HCC, and PC studies in EPIC for alcohol biomarker discovery. The authors first identified 205 features (163 in positive and 42 in negative mode) associated with alcohol intake in the CS study. These features were then manually matched by an expert to features in both the HCC and PC studies (Materials and methods, *Table 1*). In our analysis, we use these features as a validation set and compare each method's matchings to the expert manual matchings on this subset. Due to the imbalance between the number of positive and negative mode features in the validation subset, our main analysis focuses on the alignment results of CS with HCC and CS with PC in positive mode (*Table 2*). We delegate the matching results between the negative mode studies (*Table 3*) to Appendix 4.

In this section, we use the same settings for GM as in our simulation study, and do not apply an additional thresholding step. The parameters of metabCombiner and M2S are calibrated using the validation subset as prior knowledge (Appendix 2).

Preliminary analysis of the validation subset reveals inconsistencies in the mean feature intensities (*Figure 4—figure supplement 1*), but *Figure 4b* shows that on centered and scaled data, the 90 expert matched features shared between the CS and HCC studies have similar correlation structures. Hence, to avoid potential errors we center and scale the feature intensities which improves the performance of all three methods tested below (Appendix 4, *Appendix 4—table 1*).

### Hepatocellular carcinoma

Here, we analyze the quality of the matchings obtained by GM, M2S, and metabCombiner between the CS and HCC datasets in positive mode. Both GM and M2S identify approximately 1000 shared features while metabCombiner finds a smaller number of about 700 shared features. We refer the reader to *Figure 4—figure supplement 2a* for the precise matched feature sizes and details on the agreement between the feature matchings of all three methods.

We evaluate the performance of metabCombiner, M2S, and GM on the validation subset in positive mode (*Figure 4c*, *Table 2*), which consist of 90 features from the CS study manually matched to features from the HCC study and 73 features specific to the CS study. MetabCombiner demonstrates precise matching but lacks sensitivity. M2S's precision and recall are comparable with GM, in contrast to its performance on simulated data. This can be attributed to the RT drift shape between the CS and HCC studies (Appendix 2), which is estimated to be close to linear (*Figure 4—figure supplement 3*). Because the parameters of M2S are fine-tuned in the validation subset, it is able to learn this linear drift and apply tight RT thresholds to achieve accurate matchings. In contrast to metabCombiner and M2S, the GM algorithm is not given any prior knowledge of the validation subset, and nevertheless demonstrates the highest precision and recall rates of the three methods (*Figure 4c*). *Figure 4b* shows how GM recovers the majority of the expert matched pairs by leveraging the shared correlations.

### Pancreatic cancer

Matching features between the CS and PC studies in positive mode, GM and M2S identify approximately 1000 common features, while metabCombiner detects approximately 600 matches (*Figure 4—figure supplement 2b*). We examine the performance of all three methods on the validation subset consisting of 66 manually matched features between CS and PC along with 97 features specific to the CS study. As before, GM and M2S have high recall while the recall of metabCombiner is less than 0.5.

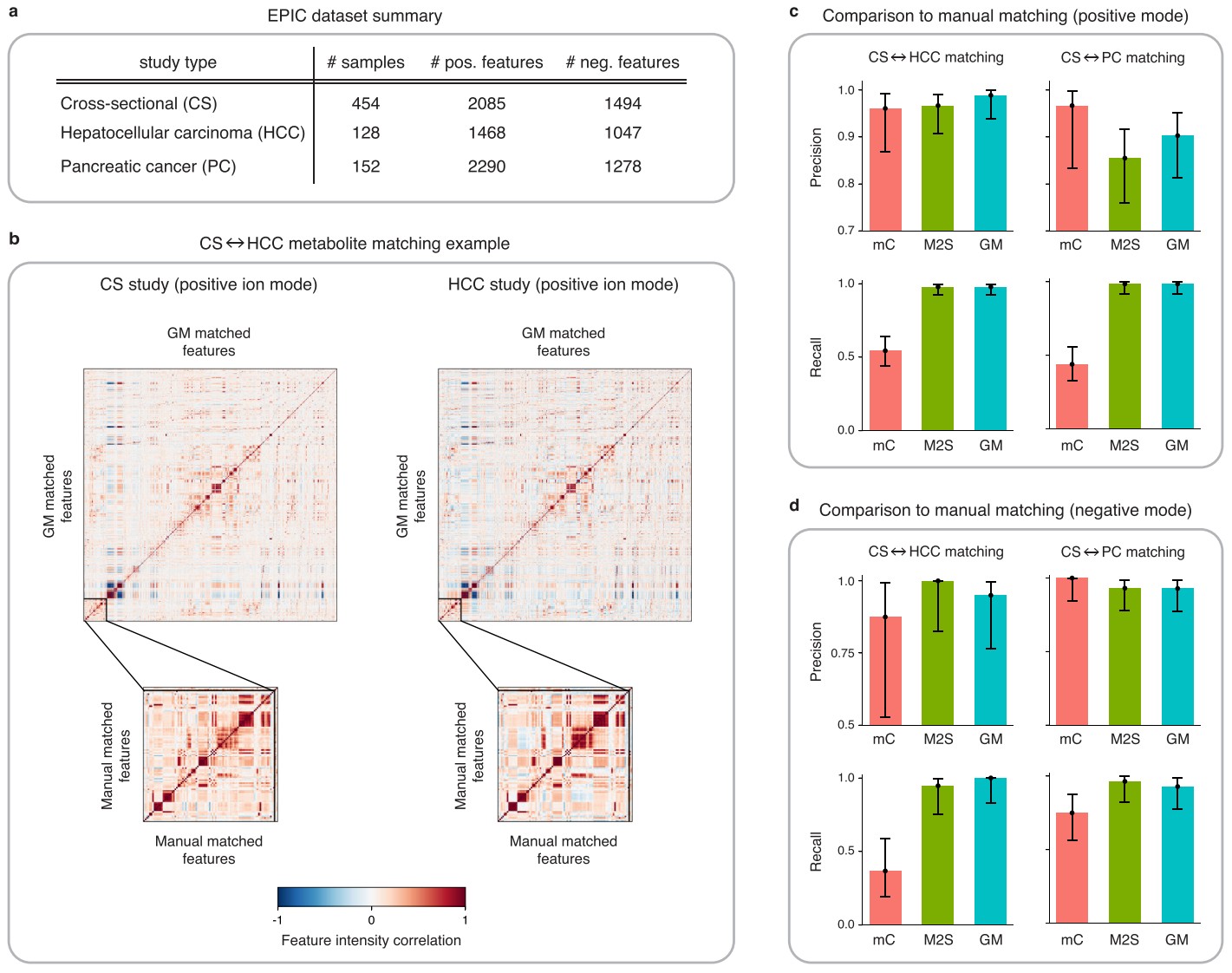

**Figure 4.** Application of GromovMatcher and comparison to existing methods on EPIC dataset. (**a**) Dimensions of the three EPIC studies used. For each ionization mode, the cross-sectional (CS) study is aligned successively with the hepatocellular carcinoma (HCC) study and the pancreatic cancer (PC) study. (**b**) Demonstration of expert manual matching and GromovMatcher (GM) matching between the CS and HCC studies in positive mode. Experts manually match 90 features (**Table 1**) from **Loftfield et al., 2021** and the correlation matrices of these features in both datasets have similar structure (bottom two matrices). GM discovers 996 shared features between the CS and HCC datasets which have similar correlation structure (top two matrices). We validate that 88 of the 90 features from the manually expert matched subset are contained in the set of features matched by GM. (**c**) Performance of metabCombiner (mC), M2S and GM in positive mode. Precisions and recalls are measured on a validation subset of 163 manually examined features, and 95% confidence intervals are computed using modified Wilson score intervals. (**d**) Performance of mC, M2S, and GM in negative mode. Precision and recall are measured on a validation subset of 42 manually examined features, and 95% confidence intervals are computed using modified Wilson score intervals. See **Table 2** and **Table 3** for exact precisions, recalls, and confidence intervals in positive and negative mode, respectively.

The online version of this article includes the following figure supplement(s) for figure 4:

**Figure supplement 1.** Consistency of the mean feature intensities (FI) in EPIC.

**Figure supplement 2.** Overlap between the matching results obtained by metabCombiner, M2S and GromovMatcher in EPIC.

**Figure supplement 3.** Estimated RT drift between the EPIC studies aligned in the main experiment.

A decrease in precision is observed for both GM and M2S compared to the previous CS-HCC matchings. We therefore manually inspect the false positive matches; the set of CS features matched by the method to the PC study but explicitly examined and left unmatched in the expert manual matching. Assessing the GM results, we identify seven false positive feature matches. Upon secondary

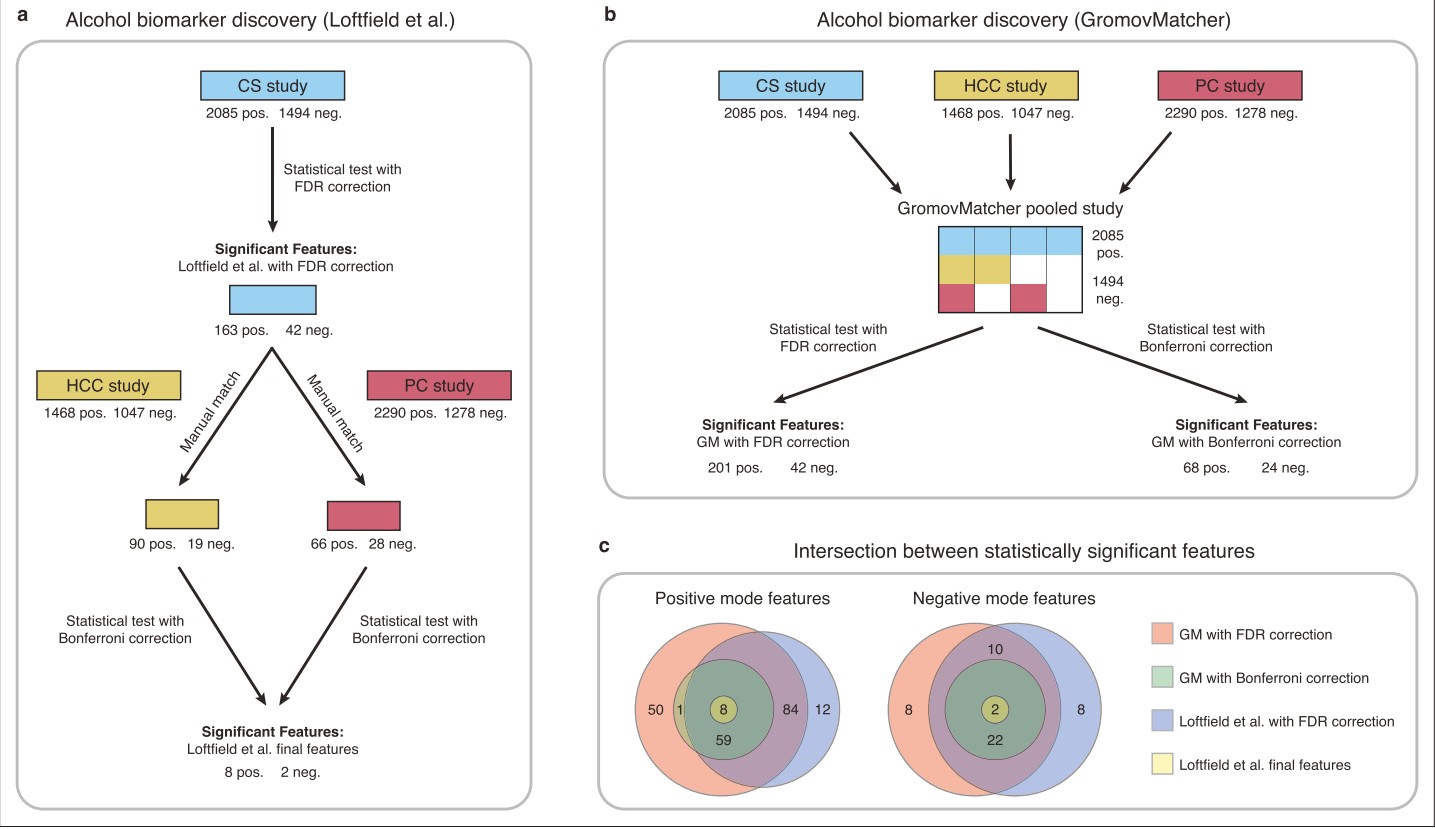

**Figure 5.** Comparison of GromovMatcher and *Loftfield et al., 2021* analysis for alcohol biomarker discovery on EPIC data. (**a**) Loftfield study implemented a discovery step, examining the relationship between alcohol intake and metabolic features in the CS study. The significant features in CS were manually matched to features from the HCC and PC and the analysis was repeated using samples from the HCC and PC studies. After this step, 10 features associated with alcohol intake were identified. (**b**) GromovMatcher analysis begins by matching features from CS study to HCC and PC studies respectively (top blue, yellow, and red boxes). Samples corresponding to each CS feature are combined with the samples of its matched feature in the HCC study, PC study, or both. This generates a larger pooled data matrix with the same number of features as the CS study but with more samples pooled across the three original studies (center matrix). Because some features in the CS study may not have matches in HCC or PC, the corresponding entries in the pooled matrix are set to NaN/missing values (white regions in matrix). Each column/feature in this matrix is statistically tested for association with alcohol intake (ignoring missing values) and an FDR or a stricter Bonferroni correction is performed to retain only a subset of features from the pooled study that have a strong association. (**c**) Venn diagrams show intersection of feature sets (in positive and negative mode) found to be associated with alcohol intake by one of the four different analyses.

inspection, three pairs are revealed as correct matches that were not initially identified in the expert matching. M2S finds 11 false positive matches which include the 7 false positives recovered by GM. Manual examination of the four remaining pairs reveals two clear mismatches. These results highlight the advantage of using automated methods for data alignment, as both GM and M2S detect correct matches that were not identified by experts, with GM being more precise than M2S.

## Illustration for alcohol biomarker discovery

*Loftfield et al., 2021* identified biomarkers of habitual alcohol intake by first performing a discovery step, where they examined the relationship between alcohol intake and metabolic features in the CS study. They then manually matched the significant features in CS to features from the HCC and PC studies, and repeated the analysis with samples from the HCC and PC studies to determine whether the association with alcohol intake persisted. This led to the identification of 10 features possibly associated with alcohol intake (*Figure 5a*).

To extend this analysis and illustrate the benefit of GM automatic matching for biomarker discovery, we use GM to pool features from the CS, HCC, and PC studies, and examine the relationship between metabolic features and alcohol intake in the pooled study (Materials and methods and *Figure 5b*).

Applying an FDR correction on the pooled study, we identify 243 features associated with alcohol intake, including 185 features consistent with the discovery step of *Loftfield et al., 2021*, and 55 newly discovered features (*Figure 5c*). Using the more stringent Bonferroni correction on the pooled data, we identify 36 features shared by all three studies that are significantly associated with alcohol intake. These features include all 10 features identified in Loftfield et al. (*Figure 5c*). These findings highlight the potential benefits of using GM automatic matching for biomarker discovery in untargeted metabolomics data. Additional information regarding the methodology and findings of our GM and Loftfield et al. analyses can be found in Materials and methods and Appendix 4.

## Discussion

LC-MS metabolomics has emerged as an increasingly powerful tool for biological and biomedical research, offering promising opportunities for epidemiological and clinical investigations. However, integrating data from different sources remains challenging. To address this issue, we introduce GromovMatcher, a method based on optimal transport that automatically aligns LC-MS data from pairs of studies. Our method exhibits superior performance on both simulated and real data when compared to existing approaches. Additionally, it presents a user-friendly interface with few hyperparameters.

While GromovMatcher is robust to noise and variations in data, it may face limitations when aligning LC-MS studies from populations with different characteristics, where the correlation structures between features may be inconsistent across studies. In this case, the base assumption of GromovMatcher can be relaxed by focusing on subsamples with similar characteristics, as exemplified in a recent study (*Gomari et al., 2022*).

A current limitation is that GromovMatcher does not account for more than two datasets simultaneously, although this can be overcome by aligning multiple studies to a chosen reference dataset, as demonstrated in our biomarker experiments. The extension of Gromov-Wasserstein to multiple distributions (*Beier et al., 2022*) is another promising approach for generalizing GromovMatcher to multiple dataset alignment. Further improvements can be made by incorporating existing knowledge about the studies being matched, such as known shared features, samples in common, or MS/MS data.

The results obtained from GromovMatcher are highly promising, opening the door for various analyses of metabolomic datasets acquired in different experimental laboratories. Here, we demonstrated the potential of GromovMatcher in expediting the combination and meta-analysis of data for biomarker and metabolic signature discovery. The matchings learned by GromovMatcher also allow for comparison between experimental protocols by assessing the drift in *m/z*, RT, and feature intensities across studies. Finally, inter-institutional annotation efforts can directly benefit from incorporating this method to transfer annotations between aligned datasets. Bridging the gap between otherwise incompatible LC-MS data, GromovMatcher enables seamless comparison of untargeted metabolomics experiments.

## Materials and methods
### GromovMatcher method overview

GromovMatcher accepts as input two feature tables from separate LC-MS untargeted metabolomics studies. Each feature table for dataset 1 and dataset 2 consists of $n_1, n_2$ biospecimen samples respectively and $p_1, p_2$ metabolic features respectively detected in the study. Features in dataset 1 are given the label $\mathrm{fx}_i$ for $i = 1, \ldots, p_1$. Every feature is characterized by a mass-to-charge ratio (*m/z*) denoted by $m_i^x$, a retention time (RT) denoted by $RT_i^x$, and a vector of intensities across all samples written as $X_i \in \mathbb{R}^{n_1}$. Similarly, features in dataset 2 are labeled as $\mathrm{fy}_j$ for $j = 1, \ldots, p_2$ and are characterized by their *m/z*, retention time $RT_j^y$, and a vector of intensities across all samples $Y_i \in \mathbb{R}^{n_2}$.

Our goal is to identify pairs of indexes $(i, j)$ with $i \in \{1, , p_1\}$ and $j \in \{1, , p_2\}$, such that $\mathrm{fx}_i$ and $\mathrm{fy}_j$ correspond to the same metabolic feature. More formally, we aim to identify a *matching matrix* $M \in \{0, 1\}^{p_1 \times p_2}$ such that $M_{ij} = 1$ if $\mathrm{fx}_i$ and $\mathrm{fy}_j$ correspond to the same feature, hereafter referred to as *matched* features. Otherwise, we set $M_{ij} = 0$.

Because the $m/z$ and RT values of metabolomic features are often noisy and subject to experimental bias, our matching algorithm leverages metabolite feature intensities $X_i, Y_j$ to produce accurate dataset alignments. The GromovMatcher method is based on the idea that signal intensities of the same metabolites measured in two different studies should exhibit similar correlation structures, in addition to having compatible $m/z$ and RT values. Here, we define the Pearson correlation for vectors $u, v \in \mathbb{R}^n$ as

$$\text{corr}(u, v) = \frac{\langle u - \bar{u}, v - \bar{v} \rangle}{\|u - \bar{u}\| \|v - \bar{v}\|} \tag{1}$$

where we define

$$\bar{u} = \frac{1}{n} \sum_{i=1}^{n} u_i, \quad \|u\| = \sqrt{\sum_{i=1}^{n} u_i^2}, \quad \langle u, v \rangle = \sum_{i=1}^{n} u_i v_i \tag{2}$$

as the mean value, Euclidean norm and inner product respectively. If measurements $X_i, Y_j$ correspond to the same underlying feature, and similarly, measurements $X_k, Y_l$ share the same an underlying feature, we expect that

$$\text{corr}(X_i, X_k) \approx \text{corr}(Y_j, Y_l). \tag{3}$$

This idea that the feature intensities of shared metabolites have the same correlation structure in both datasets also holds more generally for distances, under a suitable choice of distance. For example, the correlation coefficient $\text{corr}(u, v)$ can be turned into a dissimilarity metric by defining

$$d^{\cos}(u, v) = \sqrt{1 - \text{corr}(u, v)} \tag{4}$$

commonly referred to as the *cosine distance*. Preservation of feature intensity correlations then trivially amounts to the preservation of cosine distances.

Another classical notion of distance between vectors $u, v \in \mathbb{R}^n$ is the normalized Euclidean distance

$$d^{\text{euc}}(u, v) = \frac{1}{\sqrt{n}} \|u - v\| = \sqrt{\frac{1}{n} \sum_{i=1}^{n} (u_i - v_i)^2} \tag{5}$$

which is equal to the cosine distance (up to constants) when the vectors $u, v$ are centered and scaled to have zero mean and a standard deviation of one. The Euclidean distance depends on the magnitude or mean intensity of metabolic features, and hence is a useful metric for matching metabolites as long as these mean feature intensities are reliably collected.

To summarize, the main tenant of GromovMatcher is that if measurements $X_i, Y_j$ correspond to the same feature and $X_k, Y_l$ correspond to the same feature, then for suitably chosen distances $d_x : \mathbb{R}^{n_1} \times \mathbb{R}^{n_1} \to \mathbb{R}$ and $d_y : \mathbb{R}^{n_2} \times \mathbb{R}^{n_2} \to \mathbb{R}$, these distances are preserved

$$d_x(X_i, X_k) \approx d_y(Y_j, Y_l) \tag{6}$$

across both datasets. In this paper, the distances $d_x, d_y$ are taken to be the normalized Euclidean distances in **Equation 5**. We take care to specify those experiments where the metabolic features $X$ and $Y$ are centered and scaled. In these cases, implicitly the Euclidean distance between normalized feature vectors becomes the cosine distance **Equation 4** between the original (unnormalized) feature vectors.

## Unbalanced Gromov–Wasserstein

The goal of GromovMatcher is to learn a matching matrix $M \in \{0, 1\}^{p_1 \times p_2}$ that gives an alignment between a subset of metabolites in both datasets. However, searching over the combinatorially large set of binary matrices would be an inefficient approach for dataset alignment. The mathematical framework of optimal transport *Peyré and Cuturi, 2019* instead enlarges this space of binary matrices to the set of *coupling matrices* with real nonnegative entries $\Pi \in \mathbb{R}_+^{p_1 \times p_2}$. The entries $\Pi_{ij}$ with large

weights indicate that feature $fx_i$ in dataset 1 and feature $fy_j$ in dataset 2 are a likely match. Taking inspiration from *Equation 6*, we minimize the following objective function

$$\mathcal{E}(\Pi) = \sum_{i,k=1}^{p_1} \sum_{j,l=1}^{p_2} \Pi_{ij}\Pi_{kl}\left|d_x(X_i, X_k) - d_y(Y_j, Y_l)\right| \tag{7}$$

to estimate the coupling matrix $\Pi$.

A standard approach is to optimize this objective over all coupling matrices $\Pi$ under exact marginal constraints $\Pi\mathbf{1}_{p_2} = \frac{1}{p_1}\mathbf{1}_{p_1}, \Pi^T\mathbf{1}_{p_1} = \frac{1}{p_2}\mathbf{1}_{p_2}$. Here, we define $\mathbf{1}_n$ is the ones vector of length $n$, and $\Pi_1 = \Pi\mathbf{1}_{p_2}, \Pi_2 = \Pi^T\mathbf{1}_{p_1}$ denote the column and row sums of the coupling matrix. Objective *Equation 7* under these exact marginal constraints defines a distance between the two sets of metabolic feature vectors $\{X_i\}_{i=1}^{p_1}, \{Y_i\}_{i=1}^{p_2}$ known as the Gromov–Wasserstein distance *Mémoli, 2011*, a generalization of optimal transport to metric spaces. Note that for pairs $X_i, Y_j$ and $X_k, Y_l$ for which $d_x(X_i, X_k) \approx d_y(Y_j, Y_l)$, the entries $\Pi_{ij}, \Pi_{kl}$ are penalized less and hence matches between features $fx_i, fy_j$ and features $fx_k, fy_l$ are more favored. In our optimization, we avoid enforcing exact marginal constraints on the marginal distributions $\Pi\mathbf{1}_{p_2}$ and $\Pi^T\mathbf{1}_{p_1}$ of our coupling matrix as this would enforce that all metabolites in both datasets are matched (Appendix 1). However, without any marginal constraints on the coupling $\Pi$, the objective function *Equation 7* is trivially minimized by $\Pi = 0$, leaving all metabolites in both datasets unmatched.

To account for this, we follow the ideas of unbalanced Gromov–Wasserstein (UGW) (*Sejourne et al., 2021*) and add three regularization terms to our objective

$$\mathcal{L}_{\rho,\varepsilon}(\Pi) = \mathcal{E}(\Pi) \quad +\rho D_{\mathrm{KL}}\left(\Pi_1 \otimes \Pi_1, a \otimes a\right)$$
$$+\rho D_{\mathrm{KL}}\left(\Pi_2 \otimes \Pi_2, b \otimes b\right)$$
$$+\varepsilon D_{\mathrm{KL}}(\Pi \otimes \Pi, \left(a \otimes b\right)^{\otimes 2}) \tag{8}$$

where $\rho, \varepsilon > 0$ and we define $a = \mathbf{1}_{p_1}, b = \mathbf{1}_{p_2}$. Here $\otimes$ denotes the Kronecker product. We define $D_{\mathrm{KL}}$ as the Kullback–Leibler (KL) divergence between two discrete distributions $\mu, \nu \in \mathbb{R}^p_+$ by

$$D_{\mathrm{KL}}(\mu, \nu) = \sum_{i=1}^{p} \mu_i \ln\left(\frac{\mu_i}{\nu_i}\right) - \sum_{i=1}^{p} \mu_i + \sum_{i=1}^{p} \nu_i \tag{9}$$

which measures the closeness of probability distributions.

The first two regularization terms in *Equation 8* enforce that the row sums and column sums of the coupling matrix $\Pi$ do not deviate too much from a uniform distribution, leading our optimization to match as many metabolic features as possible. The magnitude of the regularizer $\rho$ roughly enforces the fraction of metabolites in both datasets that are matched where large $\rho$ implies most metabolites are matched across datasets. The final regularization term $\varepsilon$ in *Equation 8* controls the smoothness (entropy) of the coupling matrix $\Pi$ where larger values of $\varepsilon$ encourage $\Pi$ to put uniform weights on many of its entries, leading to less precision in the metabolite matches. However, increasing $\varepsilon$ also leads to better numerical stability and a significant speedup of the alternating minimization algorithm used to optimize the objective function (Appendix 1). In our implementation, we set $\rho$ and $\epsilon$ to the lowest possible values under which our optimization converges, with $\rho = 0.05$ and $\epsilon = 0.005$.

Our full optimization problem can now be written as

$$\mathrm{UGW}_{\rho,\varepsilon} = \min_{\Pi \in \mathbb{R}_+^{p_1 \times p_2}} \mathcal{L}_{\rho,\varepsilon}(\Pi). \tag{10}$$

The UGW objective function is optimized through alternating minimization based on the code of *Sejourne et al., 2021* using the unbalanced Sinkhorn algorithm *Séjourné et al., 2019* from optimal transport (Appendix 1).

## Constraint on *m/z* ratios

Matched metabolic features must have compatible *m/z* so we enforce that $\Pi_{ij} = 0$ when $|m_i^x - m_j^y| > m_{\mathrm{gap}}$ where $m_{\mathrm{gap}}$ is a user-specified threshold. Based on prior literature (*Loftfield et al., 2021*; *Hsu et al., 2019*; *Climaco Pinto et al., 2022*; *Habra et al., 2021*; *Chen et al., 2021*), we set $m_{\mathrm{gap}} = 0.01$ ppm.

Note that $m_{\text{gap}}$ is not explicitly used in *Equation 10* but is rather enforced in each iteration of our alternating minimization algorithm for the UGW objective (Appendix 1).

Unlike the $m/z$ ratios discussed above, RTs often exhibit a non-linear deviation (drift) between studies so we cannot enforce compatibility of RTs directly in our optimization. Instead, in the following step of our pipeline we ensure matched metabolite pairs have compatible RTs by estimating the drift function and subsequently using it to filter out metabolite matches whose RT values are inconsistent with the estimated drift.

## Estimation of the RT drift and filtering

Estimating the drift between RTs of two studies is a crucial step in assessing the validity of metabolite matches and discarding those pairs which are incompatible with the estimated drift.

Let $\tilde{\Pi} \in \mathbb{R}_+^{p_1 \times p_2}$ be the minimizer of *Equation 10* obtained after optimization. We seek to estimate the RT drift function $f : \mathbb{R}_+ \to \mathbb{R}_+$ which relates the retention times of matched features between the two studies. Namely, if feature $\text{fx}_i$ and feature $\text{fy}_j$ correspond to the same metabolic feature, then we must have that $RT_j^y \approx f(RT_i^x)$.

We propose to learn the drift $f$ through the weighted spline regression

$$\min_{f \in \mathcal{B}_{n,k}} \sum_{i=1}^{p_1} \sum_{j=1}^{p_2} \tilde{\Pi}_{ij} \left| f\left(RT_i^x\right) - RT_j^y \right| \tag{11}$$

where $\mathcal{B}_{n,k}$ is the set of $n$-order B-splines with $k$ knots. All pairs $(RT_i^x, RT_j^y)$ in objective *Equation 11* are weighted by the coefficients of $\tilde{\Pi}$ so that larger weights are given to pairs identified with high confidence in the first step of our procedure. The order of the B-splines was set to $n = 3$ by default, while the number of knots $k$ was selected by 10-fold cross-validation.

Pairs identified as incompatible with the estimated RT drift are then discarded from the coupling matrix. To do this, we first take the estimated RT drift $\hat{f}$, and the set of pairs $\mathcal{S} = \{i, j : \tilde{\Pi}_{i,j} \neq 0\}$ recovered in $\tilde{\Pi}$. We then define the residual associated with $(i, j) \in \mathcal{S}$ as

$$r_{\hat{f}}(i, j) = \left| \hat{f}(RT_i^x) - RT_j^y \right|. \tag{12}$$

The 95% prediction interval and the median absolute deviation (MAD) of these residuals are given by

$$\text{PI} = 1.96 \times \text{std}(\{r_{\hat{f}}(i, j), (i, j) \in \mathcal{S}\})$$

$$\text{MAD} = \text{median}(\{|r_{\hat{f}}(i, j) - \mu_r|, (i, j) \in \mathcal{S}\}) \tag{13}$$

$$\mu_r = \text{median}(\{|r_{\hat{f}}(i, j)|, (i, j) \in \mathcal{S}\})$$

where $|\mathcal{S}|$ is the size of $\mathcal{S}$ and the functions std and median denote the standard deviation and median respectively. Similar to the approach in *Climaco Pinto et al., 2022*, we create a new filtered coupling matrix $\widehat{\Pi} \in \mathbb{R}_+^{p_1 \times p_1}$ given by

$$\widehat{\Pi}_{ij} = \begin{cases} \tilde{\Pi}_{ij} & \text{if } r_{\hat{f}}(i, j) < \mu_r + r_{\text{thresh}} \\ 0 & \text{otherwise} \end{cases}. \tag{14}$$

where $r_{\text{thresh}}$ is a given filtering threshold. Following *Habra et al., 2021*, the estimation and outlier detection step can be repeated for multiple iterations, to remove pairs that deviate significantly from the estimated drift and improve the robustness of the drift estimation. In our main algorithm, we use two preliminary iterations where estimate the RT drift and discard outliers outside of the 95% prediction interval by setting $r_{\text{thresh}} = \text{PI}$. We the re-estimate the drift and perform a final filtering step with the more stringent MAD by setting $r_{\text{thresh}} = 2 \times \text{MAD}$.

At this stage, it is possible for $\widehat{\Pi}$ to still contain coefficients of very small magnitude. As an optional postprocessing step, we discard these coefficients by setting all entries smaller than $\tau \max(\widehat{\Pi})$ to zero, for some user-defined $\tau \in [0, 1]$. Lastly, a feature from either study could have multiple possible matches, since $\widehat{\Pi}$ can have more than one non-zero coefficient per row or column. Although reporting

multiple matches can be helpful in an exploratory context, for the sake of simplicity in our analysis, the final output of GromovMatcher returns a one-to-one matching, as we only keep those metabolite pairs $(i,j)$ where the entry $\widehat{\Pi}_{ij}$ is largest in its corresponding row and column. All nonzero entries of $\widehat{\Pi}$ which do not satisfy this criterion are set to zero. Finally, we convert $\widehat{\Pi}$ into a binary matching matrix $M \in \{0,1\}^{p_1 \times p_2}$ with ones in place of its nonzero entries and this final output is returned to the user.

As a naming convention, we use the abbreviation GM for our GromovMatcher method, and use the abbreviation GMT when running GromovMatcher with the optional $\tau$-thresholding step with $\tau = 0.3$.

## Metrics for dataset alignment

Every alignment method studied in this paper returns a binary *partial matching* matrix $M \in \{0,1\}^{p_1 \times p_1}$ which has at most one nonzero entry in each row and column. Specifically, $M_{ij} = 1$ if metabolic features $i$ and $j$ in both datasets correspond to each other and $M_{ij} = 0$ otherwise. In our simulated experiments, we compare the partial matching $M$ to a known ground-truth partial matching matrix $M^* \in \{0,1\}^{p_1 \times p_2}$.

To do this, we first compute the number of true positives, false positives, true negatives, and false negatives as

$$
\begin{aligned}
\text{TP} &= \sum_{i=1}^{p_1} \sum_{j=1}^{p_2} \mathbf{1}_{M_{ij}=1} \mathbf{1}_{M_{ij}^*=1} \\
\text{FP} &= \sum_{i=1}^{p_1} \sum_{j=1}^{p_2} \mathbf{1}_{M_{ij}=1} \mathbf{1}_{M_{ij}^*=0} \\
\text{TN} &= \sum_{i=1}^{p_1} \sum_{j=1}^{p_2} \mathbf{1}_{M_{ij}=0} \mathbf{1}_{M_{ij}^*=0} \\
\text{FN} &= \sum_{i=1}^{p_1} \sum_{j=1}^{p_2} \mathbf{1}_{M_{ij}=0} \mathbf{1}_{M_{ij}^*=1}
\end{aligned}
\tag{15}
$$

where $\mathbf{1}$ denotes the indicator function. Then we use these values to compute the precision and recall as

$$
\begin{aligned}
\text{Precision} &= \frac{\text{TP}}{\text{TP} + \text{FP}} \\
\text{Recall} &= \frac{\text{TP}}{\text{TP} + \text{FN}}.
\end{aligned}
\tag{16}
$$

Precision measures the fraction of correctly found matches out of all discovered metabolite matches, while recall, also know as sensitivity, measures the fraction of correctly matched pairs out of all truly matched pairs. These two statistics can be summarized into one metric called the F1-score by taking their harmonic mean

$$
\text{F1} = 2 \cdot \frac{\text{Precision} \cdot \text{Recall}}{\text{Precision} + \text{Recall}}
\tag{17}
$$

These three metrics, precision, recall, and the F1-score, are used throughout the paper to assess the performance of dataset alignment methods, both on simulated data where the ground-truth matching is known, and on the validation subset in EPIC, using results from the manual examination as the ground-truth benchmark.

## Validation on simulated data

To assess the performance of GromovMatcher and compare it to existing dataset alignment methods, we simulate realistic pairs of untargeted metabolomics feature with known ground-truth matchings. This allows us to analyze the dependence of alignment methods on the number of shared metabolites, dataset noise level, and feature intensity centering and scaling.

### Dataset generation

Our pairs of synthetic feature tables are generated from one real untargeted metabolomics study of 500 newborns within the EXPOsOMICS project, which uses reversed phase liquid chromatography-quadrupole time-of-flight mass spectrometry (UHPLC-QTOF-MS) system in positive ion mode *Alfano et al., 2020*. The original dataset is first preprocessed following the procedure detailed in *Alfano*

*et al., 2020*, resulting in p=4712 features measured in $n = 499$ samples available for subsequent analysis. Features and samples from the original study are then divided into two feature tables of respective size $(n_1, p_1)$ and $(n_2, p_2)$, with $n_1 + n_2 = n$ and $p_1, p_2 \leq p$. In order to do this, $n_1 = \lfloor n/2 \rfloor$ randomly chosen samples from the original study are placed into dataset 1 and the remaining $n_2 = \lceil n/2 \rceil$ samples from the original study are placed into dataset 2. Here, $\lfloor \cdot \rfloor$ and $\lceil \cdot \rceil$ denote integer floor and ceiling functions. The features of the original study are randomly assigned to dataset 1, dataset 2, or both, allowing the resulting studies to have both common and study-specific features (*Figure 2*). Specifically, for a fixed overlap parameter $\lambda \in [0, 1]$, we assign a random subset of $\approx \lambda p$ features into both dataset 1 and dataset 2 while the remaining $\approx (1 - \lambda p)$ features are divided equally between the two studies such that $p_1 = p_2$. We choose $\lambda \in \{0.25, 0.5, 0.75\}$ corresponding to low, medium and high overlap. For more detailed information on how the dataset split is performed and for additional validation experiments with unbalanced dataset splits (e.g. $n_1 \neq n_2, p_1 \neq p_2$) we refer the reader to Appendix 3.

After generating a pair of studies, random noise is added to the *m/z*, RT and intensity levels of features in dataset 2 to mimic variations in data acquisition across two different experiments. The noise added to each *m/z* value in study **2** is sampled from a uniform distribution on the interval $[-\sigma_M, \sigma_M]$ with $\sigma_M = 0.01$ (*Climaco Pinto et al., 2022*). The RTs of dataset 2 are first deviated by the function $f(x) = 1.1x + 1.3\sin(1.2\sqrt{x})$, corresponding to a systematic inter-dataset drift (*Habra et al., 2021*; *Climaco Pinto et al., 2022*; *Brunius et al., 2016*). A uniformly distributed noise on the interval $[-\sigma_{RT}, \sigma_{RT}]$ is added to the deviated RTs of dataset 2, with $\sigma_{RT} \in \{0.2, 0.5, 1\}$ (in minutes) corresponding to low, moderate and high variations (*Climaco Pinto et al., 2022*; *Habra et al., 2021*; *Vaughan et al., 2012*). Finally, we add a Gaussian noise $\mathcal{N}(0, \sigma_{FI}^2)$ to the feature intensities of both studies where $\sigma_{FI}$ is the scalar variance of the noise. This noise perturbs the correlation matrices of dataset 1 and dataset 2, making matching based on feature intensity correlations more challenging. We vary $\sigma_{FI}$ over the set of values {0.1, 0.5, 1}.

Given this data generation process, we test the performance of the four alignment methods (M2S, metabCombiner, GM, and GMT) under the parameter settings described below.

## Dependence on overlap

We first assess how the performance of the four methods is affected by the number of metabolic features shared in both datasets. For each value of $\lambda = 0.25, 0.5, 0.75$ (low, medium, and high overlap), we randomly generate 20 pairs of datasets with noise on the *m/z*, RT and feature intensities set to $\sigma_M = 0.01, \sigma_{RT} = 0.5, \sigma_{FI} = 0.5$. The precision and recall of each method at low, medium, and high overlap is recorded for each of the repetitions.

## Noise robustness

Next, we test the robustness to noise of each method by fixing the metabolite overlap fraction at $\lambda = 0.5$ and generating 20 random pairs of datasets at low ($\sigma_{RT} = 0.2, \sigma_{FI} = 0.1$), medium ($\sigma_{RT} = 0.5, \sigma_{FI} = 0.5$), and high ($\sigma_{RT} = 1, \sigma_{FI} = 1$) noise levels. Similarly, the precision and recall of each method is saved for each noise level across the 20 repetitions.

## Feature intensity centering and scaling

In order to test how all four methods are affected when the mean feature intensities and variance are not comparable across studies, we assess their performance when the feature intensities in both studies are mean centered and standardized to have unit standard deviation across all samples. We again generate 20 random pairs of datasets with medium overlap and medium noise, normalize the

**Table 1.** Results from the manual matching conducted for *Loftfield et al., 2021*.
Features from the CS study (163 features in positive mode, 42 features in negative mode) were manually investigated for matches in the HCC and PC studies.

| Study | Manual matches found in positive mode | Manual matches found in negative mode |
|---|---|---|
| Hepatocellular carcinoma (HCC) | 90 | 19 |
| Pancreatic cancer (PC) | 66 | 28 |

**Table 2.** Precision and recall on the EPIC validation subset in positive mode.
95% confidence intervals were computed using modified Wilson score intervals (*Brown et al., 2001*; *Agresti and Coull, 1998*).

| | CS ⟷ HCC | | CS ⟷ PC | |
|---|---|---|---|---|
| Method | Precision | Recall | Precision | Recall |
| GromovMatcher | 0.989 (0.939, 0.999) | 0.978 (0.923, 0.996) | 0.903 (0.813, 0.952) | 0.985 (0.919, 0.999) |
| M2S | 0.967 (0.908, 0.991) | 0.978 (0.923, 0.996) | 0.855 (0.759, 0.917) | 0.985 (0.919, 0.999) |
| metabCombiner | 0.961 (0.868, 0.993) | 0.544 (0.442, 0.643) | 0.967 (0.833, 0.998) | 0.439 (0.326, 0.559) |

**Table 3.** Precision and recall on the EPIC validation subset in negative mode.
95% confidence intervals were computed using modified Wilson score intervals (*Brown et al., 2001*; *Agresti and Coull, 1998*).

| | CS ⟷ HCC | | CS ⟷ PC | |
|---|---|---|---|---|
| Method | Precision | Recall | Precision | Recall |
| GromovMatcher | 0.950 (0.764, 0.997) | 1.000 (0.832, 1.000) | 0.929 (0.774, 0.987) | 0.929 (0.774, 0.987) |
| M2S | 1.000 (0.824, 1.000) | 0.947 (0.754, 0.997) | 0.931 (0.780, 0.988) | 0.964 (0.823, 0.998) |
| metabCombiner | 0.875 (0.529, 0.993) | 0.368 (0.191, 0.590) | 1.000 (0.845, 1.000) | 0.750 (0.566, 0.873) |

feature intensities in each pair of datasets, and compute the precision and recall of each method across the 20 repetitions.

## EPIC data

We also evaluate our method on data collected within the European Prospective Investigation into Cancer and Nutrition (EPIC) cohort, an ongoing multicentric prospective study with over 500,000 participants recruited between 1992 and 2000 from 23 centers in 10 European countries, and who provided blood samples at the inclusion in the study (*Riboli et al., 2002*). In EPIC, untargeted metabolomics data were successively acquired in several studies nested within the full cohort.

In the present work, we use untargeted metabolomics data acquired in three studies nested in EPIC, namely the EPIC cross-sectional (CS) study (*Slimani et al., 2003*) and two matched case-control studies nested within EPIC, on hepatocellular carcinoma (HCC; *Stepien et al., 2016*; *Stepien et al., 2021*) and pancreatic cancer (PC; *Gasull et al., 2019*), respectively. All data were acquired at the International Agency for Research on Cancer, making use of the same plateform and methodology: UHPLC-QTOF-MS (1290 Binary Liquid chromatography system, 6550 quadrupole time-of-flight mass spectrometer, Agilent Technologies, Santa Clara, CA) using reversed phase chromatography and electrospray ionization in both positive and negative ionization mode.

In a previous analysis aiming at identifying biomarkers of habitual alcohol intake in EPIC, the 205 features associated with alcohol intake in the CS study were manually matched to features in both the HCC and PC studies *Loftfield et al., 2021*. The results from this manual matching are presented in *Table 1*. This matching process was based on the proximity of $m/z$ and RT, using a matching tolerance of ± 15 ppm and ± 0.2 min, and on the comparison of the chromatograms of features in a quality control samples from both studies.

### Preprocessing

In the HCC and PC studies, samples corresponding to participants selected as cases in either study (i.e. participants selected in the study because of a diagnosis of incident HCC or PC) are excluded. Indeed, the metabolic profiles of participants selected as controls are expected to be more comparable across studies than those of cases, especially if certain features are associated with the risk of HCC or PC. Apart from this additional exclusion criterion, the untargeted metabolomics data of each study is pre-processed following the steps described in *Loftfield et al., 2021*, to eliminate unreliable features and samples, impute missing values and minimize technical variations in the feature intensity levels.

## Alcohol biomarker discovery

*Loftfield et al., 2021* used the untargeted metabolomics data of the CS, HCC and PC studies in their alcohol biomarker discovery study in EPIC, without being able to automatically match their common features and pool the three datasets. Instead, the authors first implemented a discovery step, examining the relationship between alcohol intake and metabolic features measured in the CS study and accounting for multiple testing using a false discovery rate (FDR) correction. This led to the identification of 205 features significantly associated with alcohol intake in the CS study. In order to gauge the robustness of these associations, the authors of *Loftfield et al., 2021* then implemented a validation step using data from two independent test sets. The first test set was composed of data from the EPIC HCC and PC studies, while the second was derived from the Finnish Alpha-Tocopherol, Beta-Carotene Cancer Prevention (ATBC) study. The 205 features identified in the discovery step were manually investigated for matches in the EPIC test set, and 67 features were effectively matched to features in the HCC or PC study, or both. The authors then evaluated the association between alcohol intake and those 67 features, applying a more conservative Bonferroni correction to determine whether the association with alcohol intake persisted. This step led to the identification of 10 features associated with alcohol intake (Extended Data *Figure 5a*). The second test set was then used to determine whether those 10 features were also significant in the ATBC population, which was indeed the case.

To conduct a more in-depth investigation of the matchings produced by the GromovMatcher algorithm, we build upon the analysis previously conducted by *Loftfield et al., 2021* by exploring potential alcohol biomarkers using a pooled dataset created from the CS, HCC, and PC studies. Our goal is to assess whether pooling the data leads to increased statistical power and allows for the detection of more features associated with alcohol intake. Namely, we generate the pooled dataset by aligning a chosen reference dataset (CS study) with the HCC and PC studies successively using the GM matchings computed in both positive and negative mode (Materials and methods and Extended Data *Figure 5b*). Features that are not detected in either the HCC or PC studies are designated as 'missing' in the final pooled dataset for samples belonging to the respective studies where the feature is not found.

To evaluate the potential relationship between alcohol consumption and pooled metabolic features, we use a methodology akin to that of *Loftfield et al., 2021*. The self-reported alcohol intake data is adjusted for various demographic and lifestyle factors (age, sex, country, body-mass-index, smoking status and intensity, coffee consumption, and study) via the residual method in linear regression models. Feature intensities are also adjusted for technical variables (plate number and position within the plate) via linear mixed effect models. The significance of the association is assessed using correlation coefficients computed from the residuals for both self-reported alcohol intake and feature intensities. p-Values are corrected using either false discovery rate (FDR) or Bonferroni correction to account for multiple testing. Corrected p-values less than 5% are considered significant.

## Materials and correspondence

All correspondence and material requests should be addressed to V.V.

## IARC disclaimer

Where authors are identified as personnel of the International Agency for Research on Cancer/World Health Organization, the authors alone are responsible for the views expressed in this article and they do not necessarily represent the decisions, policy, or views of the International Agency for Research on Cancer/World Health Organization.

## Acknowledgements

We thank Jörn Dunkel for helpful advice on our manuscript. We acknowledge the MIT SuperCloud and Lincoln Laboratory Supercomputing Center *Reuther et al., 2018* for providing HPC resources that have contributed to the research results reported within this paper. GS acknowledges support through a National Science Foundation Graduate Research Fellowship under Grant No. 1745302. PR is supported by NSF grants IIS-1838071, DMS-2022448, and CCF-2106377. MB and VV acknowledge support from World Cancer Research Fund (UK) through the World Cancer Research Fund International

grant program (grant number: IIG_FULL_2022_013). We are grateful to the Principal Investigators of each of the EPIC centers for sharing the data for our experimental application.

## Additional information

### Funding

| Funder | Grant reference number | Author |
|---|---|---|
| National Science Foundation | Graduate Research Fellowship Program 1745302 | George Stepaniants |
| National Science Foundation | IIS-1838071 | Philippe Rigollet |
| National Science Foundation | DMS-2022448 | Philippe Rigollet |
| National Science Foundation | CCF-2106377 | Philippe Rigollet |
| World Cancer Research Fund International | IIG_FULL_2022_013 | Marie Breeur Vivian Viallon |

The funders had no role in study design, data collection and interpretation, or the decision to submit the work for publication.

### Author contributions

Marie Breeur, George Stepaniants, Software, Formal analysis, Validation, Investigation, Visualization, Methodology, Writing - original draft, Writing - review and editing; Pekka Keski-Rahkonen, Data curation, Writing - original draft, Writing - review and editing; Philippe Rigollet, Vivian Viallon, Conceptualization, Resources, Supervision, Funding acquisition, Investigation, Methodology, Writing - original draft, Writing - review and editing

### Author ORCIDs

Marie Breeur  http://orcid.org/0000-0003-1251-8360
George Stepaniants  http://orcid.org/0000-0002-7834-7536
Pekka Keski-Rahkonen  http://orcid.org/0000-0001-9437-3040
Vivian Viallon  http://orcid.org/0000-0002-9799-4421

### Ethics

The EPIC study, and in particular the three studies nested within EPIC, were conducted according to the Declaration of Helsinki and approved by the ethics committee at the International Agency for Research on Cancer (IARC) (IEC 10-16 for the HCC and pancreatic cancer studies, IEC 12-29 for the cross-sectional study). Written informed consent was obtained from all subjects involved in the study.

Reviewer #1 (Public Review): https://doi.org/10.7554/eLife.91597.3.sa1
Reviewer #2 (Public Review): https://doi.org/10.7554/eLife.91597.3.sa2
Author response https://doi.org/10.7554/eLife.91597.3.sa3

## Additional files

### Supplementary files
• MDAR checklist

### Data availability

The LC-MS data used to generate our simulated validation experiments can be downloaded at https://www.ebi.ac.uk/metabolights/MTBLS1684/files at the bottom of the "Files" section in under filename 'FILES/metabolomics\_normalized\_data.xlsx'. The EPIC data is considered sensitive data

and is therefore not publicly available. It is centralised at IARC and can be analysed through the IARC Scientific IT platform after a Data Use Agreement has been signed. Access requests should be submitted to the IARC Steering Committee https://epic.iarc.fr/access/index.php. All code for the data preprocessing, figure generation, as well as the GromovMatcher algorithm and its comparison to other methods are available at: https://github.com/sgstepaniants/GromovMatcher (copy archived at *Breeur and Stepaniants, 2024*). Instructions and examples for how to run the GromovMatcher method are provided in the Github repository. The metabCombiner implementation written by the original authors was taken from their Github codebase: https://github.com/hhabra/metabCombiner (*Habra, 2024*). The M2S implementation of the original authors was taken from their Github codebase: https://github.com/rjdossan/M2S (*Rjdossan, 2024*).

The following previously published dataset was used:

| Author(s) | Year | Dataset title | Dataset URL | Database and Identifier |
|---|---|---|---|---|
| Vineis P, Alfano R, Chadeau-Hyam M, Keski-Rahkonen P, Robinot N, Scalbert A, Robinson O, Plusquin M | 2020 | A multi-omic analysis of birthweight in newborn cord blood reveals new underlying mechanisms related to cholesterol metabolism | https://www.ebi.ac.uk/metabolights/editor/MTBLS1684/descriptors | MetaboLights, MTBLS1684 |

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

## Appendix 1

In this paper, we study how to match metabolic features across two datasets where Dataset 1 has $p_1$ metabolic features measured across $n_1$ patients and Dataset 2 has $p_2$ metabolic features measured across $n_2$ patients. Our goal is to identify pairs of indexes $(i,j)$ with $i \in \{1, \ldots, p_1\}$ and $j \in \{1, \ldots, p_2\}$, such that feature $i$ in Dataset 1 and feature $j$ in Dataset 2 correspond to the same metabolic feature. More formally, we aim to identify a *matching matrix* $M^* \in \{0,1\}^{p_1 \times p_2}$ such that $M^*_{i,j} = 1$ if features $i$ in Dataset 1 and feature $j$ in Dataset 2 correspond to the same feature, hereafter referred to as *matched* features. Otherwise we set $M^*_{i,j} = 0$ otherwise. We emphasize that a matching matrix $M^*$ can have at most one nonzero entry in each row and column.

Both of the datasets we aim to match are obtained from liquid chromatography-mass spectrometry (LC-MS) experiments. Hence, for Dataset 1 each metabolite $i \in [p_1]$ is labeled with a mass-to-charge (*m/z*) ratio $m_i^1$ as well as a retention time (RT) given by $RT_i^1$. Additionally, each metabolite has a vector of intensities across patients denoted by $X_i \in \mathbb{R}^{n_1}$. Similarly, each metabolite $j \in [p_2]$ in Dataset 2 is labeled by its *m/z* ratio $m_j^2$, its retention time $RT_j^2$ and its vector of intensities across samples $Y_j \in \mathbb{R}^{n_2}$.

### Correlations and distances between metabolomic features

Features cannot be aligned based on their *m/z* and RT alone as they are often too inconsistent across studies. Our method is based on the idea that, in addition to their *m/z* and RT being compatible, the signal intensities of metabolites measured in two different studies should exhibit similar correlation structures, or more generally exhibit similar distances between their intensity vectors. In other words, if feature intensity vectors $X_i \in \mathbb{R}^{n_1}, Y_j \in \mathbb{R}^{n_2}$ correspond to the same underlying feature ($M^*_{ij} = 1$) and similarly if $X_k \in \mathbb{R}^{n_1}, Y_l \in \mathbb{R}^{n_2}$ correspond to the same feature ($M^*_{kl} = 1$), then we expect that

$$\text{corr}(X_i, X_k) \approx \text{corr}(Y_j, Y_l) \quad \text{if } \Pi^*_{ij} = \Pi^*_{kl} = 1. \tag{18}$$

Here we define $\text{corr}(u, v)$ to be the Pearson correlation coefficient between two feature intensity vectors $u, v \in \mathbb{R}^n$ by

$$\text{corr}(u, v) = \frac{\langle u - \bar{u}, v - \bar{v} \rangle}{\|u - \bar{u}\| \|v - \bar{v}\|} \tag{19}$$

where we define

$$\bar{u} = \frac{1}{n} \sum_{i=1}^{n} u_i, \quad \|u\| = \sqrt{\sum_{i=1}^{n} u_i^2}, \quad \langle u, v \rangle = \sum_{i=1}^{n} u_i v_i \tag{20}$$

as the mean value, Euclidean norm and inner product respectively. More generally, with $d_x$ and $d_y$ denoting two given distances on $\mathbb{R}^{n_1}$ and $\mathbb{R}^{n_2}$ respectively, we expect that

$$d_x(X_i, X_k) \approx d_y(Y_j, Y_l) \quad \text{if } \Pi^*_{ij} = \Pi^*_{kl} = 1. \tag{21}$$

Throughout this paper, we use the normalized Euclidean distance defined for any $u, v \in \mathbb{R}^n$ as

$$d^{\text{euc}}(u, v) = \frac{1}{\sqrt{n}} \|u - v\| \tag{22}$$

where for $d_x$ and $d_y$ we take $n = n_1, n_2$ respectively. If the signal intensity vectors $u, v$ are mean centered and normalized by their standard deviation as

$$u \mapsto \sqrt{n} \cdot \frac{u - \frac{1}{n} \sum_{i=1}^{n} u_i}{\left\| u - \frac{1}{n} \sum_{i=1}^{n} u_i \right\|} \tag{23}$$

and likewise for $v$, then it follows that

$$d^{\text{euc}}(u, v) = \sqrt{2\left(1 - \text{corr}(u, v)\right)} = \sqrt{2} d^{\text{cos}}(u, v) \tag{24}$$

where we denote $d^{\cos}(u, v) = \sqrt{1 - \mathrm{corr}(u, v)}$ as the cosine distance. For the purposes of this paper, we will always assume that $d_x$ and $d_y$ denote the normalized Euclidean distance from **Equation 22**. As shown above, this will be implicitly equal to the cosine distance from **Equation 24** on centered and scaled data.

The goal of metabolomic feature matching is to learn the binary matching matrix $M^*$ that aligns the distances between pairs of features in the most consistent way possible as shown in **Equation 21**. To formalize this notion into a practical algorithm, we use the mathematical theory of optimal transport (**Peyré and Cuturi, 2019**) which we discuss next.

## Optimal transport

Optimal transport (OT) applies in the setting when the points $\{X_i\}_{i=1}^{p_1}$ and $\{Y_j\}_{j=1}^{p_2}$ being matched live in the same dimensional space $n_1 = n_2 = n$. It aims to find a matching between each point $X_i$ and its corresponding point $Y_j$ such that the sum of distances between matches is minimized. Matches between each pair of points can be stored in a matching matrix $M \in \{0, 1\}^{p_1 \times p_2}$ such that $M_{ij} = 1$ if $X_i$ and $Y_j$ are matched, and $M_{ij} = 0$ otherwise. Again we note that $M$ must have at most one nonzero entry in each row and column to be a valid matching matrix.

Instead of searching over this space of binary matching matrices, optimal transport places masses $a_i \geq 0$ at all points $X_i$ for $i = 1, \ldots, p_1$ and masses $b_j \geq 0$ at all points $Y_j$ for $j = 1, \ldots, p_2$ and optimizes over the space of probabilistic *couplings* $\Pi \in \mathbb{R}_+^{p_1 \times p_2}$ which move a $\Pi_{ij}$ amount of mass from $X_i$ to $Y_j$. We assume here for simplicity that the sum of masses in both datasets are equal to one $\sum_{i=1}^{p_1} a_i = \sum_{i=1}^{p_2} b_i = 1$ and that the coupling $\Pi$ transports all mass from $a$ into $b$. More formally, optimal transport optimizes over the constrained set of couplings

$$\mathbf{U}(a, b) = \left\{ \Pi \in \mathbb{R}_+^{p_1 \times p_2} : \Pi \mathbf{1}_{p_2} = a \text{ and } \Pi^T \mathbf{1}_{p_1} = b \right\} \tag{25}$$

where $\mathbf{1}_p$ denotes the all ones vector of length $p$. In practice, the points $X_i$ and $Y_j$ in each dataset are all treated the same and the masses placed on the data are chosen to be uniform $a = \frac{1}{p_1} \mathbf{1}_{p_1}$ and $b = \frac{1}{p_2} \mathbf{1}_{p_2}$.

The cost function which optimal transport minimizes is the sum of squared distances of its transported mass

$$\mathcal{E}(\Pi) = \sum_{i=1}^{p_1} \sum_{j=1}^{p_2} \Pi_{ij} d(X_i, Y_j) \tag{26}$$

where $d(u, v) = d^{\mathrm{euc}}(u, v) = \frac{1}{\sqrt{n}} \|u - v\|$ is the Euclidean distance. The distance matrix $d(X_i, Y_j)$ in the OT objective can be replaced more generally with a cost matrix $C \in \mathbb{R}^{p_1 \times p_2}$ that is not necessarily a distance matrix. In this case the cost function becomes

$$\mathcal{E}(\Pi) = \sum_{i=1}^{p_1} \sum_{j=1}^{p_2} \Pi_{ij} C_{ij} \tag{27}$$

When the transport cost $C_{ij}$ is a distance, the OT optimization defines a valid distance metric known as the *optimal transport distance* between discrete distributions $\{(a_i, X_i)\}_{i=1}^{p_1}$ and $\{(b_j, Y_j)\}_{j=1}^{p_2}$ in $\mathbb{R}^n$ given by

$$\mathrm{OT}(a, b) = \min_{\Pi \in \mathbf{U}(a,b)} \mathcal{E}(\Pi). \tag{28}$$

When $d(u, v)$ is Euclidean, this OT distance is also referred to as the $L^1$ optimal transport distance, the Wasserstein 1-distance, or the Earth mover's distance. As formulated, the computation of the optimal transport objective involves an optimization over coupling matrices $\Pi$ which can be solved by linear programming (**Peyré and Cuturi, 2019**). The OT optimization problem becomes time consuming for problems with many points $p_1, p_2 \gg 1$. We show in the next section how augmenting this distance with a regularization term leads to a more efficient algorithm for learning the optimal coupling $\Pi$.

## Entropic regularization

Define the Kullback–Leibler (KL) divergence between two positive vectors $\mu, \nu \in \mathbb{R}^p_+$ as

$$D_{\text{KL}}(\mu, \nu) = \sum_{i=1}^{p} \mu_i \ln \left( \frac{\mu_i}{\nu_i} \right) - \sum_{i=1}^{p} \mu_i + \sum_{i=1}^{p} \nu_i \tag{29}$$

Given fixed marginals $a \in \mathbb{R}^{p_1}$ and $b \in \mathbb{R}^{p_2}$ from the previous section, we can define the entropy of a coupling matrix $\Pi \in \mathbb{R}^{p_1 \times p_2}_+$ with respect to these fixed marginals as

$$D_{\text{KL}}(\Pi, a \otimes b) = \sum_{i=1}^{p_1} \sum_{j=1}^{p_2} \Pi_{ij} \ln \left( \frac{\Pi_{ij}}{a_i b_j} \right) - \sum_{i=1}^{p_1} \sum_{j=1}^{p_2} \Pi_{ij} + \sum_{i=1}^{p_1} \sum_{j=1}^{p_2} a_i b_j. \tag{30}$$

where $(a \otimes b)_{ij} = a_i b_j$ denotes the outer product. This can be further simplified as

$$
\begin{aligned}
D_{\text{KL}}(\Pi, a \otimes b) &= \sum_{i=1}^{p_1} \sum_{j=1}^{p_2} \Pi_{ij} \ln \left( \frac{\Pi_{ij}}{a_i b_j} \right) - \sum_{i=1}^{p_1} \sum_{j=1}^{p_2} \Pi_{ij} + \left( \sum_{i=1}^{p_1} a_i \right) \left( \sum_{j=1}^{p_2} b_j \right) \\
&= \sum_{i=1}^{P_1} \sum_{j=1}^{P_2} \Pi_{ij} \ln \left( \frac{\Pi_{ij}}{a_i b_j} \right) \\
&= H(\Pi) + \ln(p_1) + \ln(p_2)
\end{aligned}
\tag{31}
$$

where we define $H(\Pi)$ by

$$H(\Pi) = \sum_{i=1}^{p_1} \sum_{j=1}^{p_2} \Pi_{ij} \ln(\Pi_{ij}). \tag{32}$$

In the second line of the derivation above, we used the fact that the entries of $a, b$, and $\Pi$ summed to one, and in the third line we used the fact that the marginals $a$ and $b$ were uniform. Under these assumptions, we see that the KL divergence $D_{\text{KL}}(\Pi, a \otimes b)$ is independent of the values of the marginals $a, b$ and is equal to $H(\Pi)$ up to constants.

Although here the general definition of entropy through the KL divergence reduces to the simpler formula of $H(\Pi)$, in the following sections we will need to extend our analysis to cases when $a, b$, and $\Pi$ have positive values that do not sum to one (i.e. not distributions). In this context, we will no longer have that $D_{\text{KL}}(\Pi, a \otimes b) = H(\Pi) + const$ but we will still be able to use $D_{\text{KL}}(\Pi, a \otimes b)$ as a general notion of entropy for $\Pi$.

The entropy of a coupling $D_{\text{KL}}(\Pi, a \otimes b)$ is an important notion because it quantifies how uniform or smooth $\Pi$ is with respect to the product distribution $a \otimes b$. In particular, if $a$ and $b$ are set to uniform distributions as commonly done in practice, then $D_{\text{KL}}(\Pi, a \otimes b)$ is small when $\Pi$ has close to uniform entries and is large otherwise. This notion of smoothness allows us to use $D_{\text{KL}}(\Pi, a \otimes b)$ as a regularizer in our optimal transport distance as

$$\mathcal{L}_\varepsilon(\Pi) = \sum_{i=1}^{p_1} \sum_{j=1}^{p_2} \Pi_{ij} C_{ij} + \varepsilon D_{\text{KL}}(\Pi, a \otimes b) \tag{33}$$

where $\varepsilon$ is a small regularization parameter. Note that here we have denoted the transport cost matrix by $C \in \mathbb{R}^{p_1 \times p_2}$ which is not necessarily a distance matrix. The introduction of the regularizer $\varepsilon D_{\text{KL}}(\Pi, a \otimes b)$ gives us an efficient iterative algorithm known as the *Sinkhorn algorithm* for optimizing $\Pi$ which we describe in the following sections.

## Unbalanced optimal transport

Before we introduce the Sinkhorn algorithm, we introduce a final modification to our optimal transport distance that allows us to learn couplings between distributions $a, b \in \mathbb{R}^p_+$ that do not preserve mass. In other words, the coupling $\Pi$ is not required to perfectly satisfy the marginal constraints $\Pi \mathbf{1}_{p_2} = a$ and $\Pi^T \mathbf{1}_{p_1} = b$. In our metabolite matching problem, this is particularly useful as not all metabolites in one dataset necessarily appear in the other dataset and hence should be left unmatched. This modification of optimal transport, known as unbalanced optimal transport (UOT) *Chizat et al., 2018*, optimizes the following cost function

$$\mathcal{L}_{\rho,\varepsilon}(\Pi) = \sum_{i=1}^{p_1} \sum_{j=1}^{p_2} \Pi_{ij} C_{ij} + \rho D_{\mathrm{KL}}(\Pi \mathbf{1}_{p_2}, a) + \rho D_{\mathrm{KL}}(\Pi^T \mathbf{1}_{p_1}, b) + \varepsilon D_{\mathrm{KL}}(\Pi, a \otimes b) \tag{34}$$

where we have added two KL terms with regularization parameter $\rho$ to enforce that the marginals of the coupling $\Pi \mathbf{1}_{p_2}, \Pi^T \mathbf{1}_{p_1}$ are approximately close to the prescribed marginals $a, b$ respectively. We have also kept the smoothness/entropy regularizer $\varepsilon D_{\mathrm{KL}}(\Pi, a \otimes b)$ from the previous section.

## Unbalanced Sinkhorn algorithm

Now we are ready to present the unbalanced Sinkhorn algorithm *Peyré and Cuturi, 2019* for optimizing the unbalanced optimal transport cost defined above. First we rewrite our optimization as

$$
\begin{aligned}
\min_{\Pi \in \mathbb{R}_+^{p_1 \times p_2}} \mathcal{L}_{\rho,\varepsilon}(\Pi) &= \min_{\Pi \in \mathbb{R}_+^{p_1 \times p_2}} \sum_{i=1}^{p_1} \sum_{j=1}^{p_2} \Pi_{ij} C_{ij} + \rho D_{\mathrm{KL}}(\Pi \mathbf{1}_{p_2}, a) + \rho D_{\mathrm{KL}}(\Pi^T \mathbf{1}_{p_1}, b) + \varepsilon D_{\mathrm{KL}}(\Pi, a \otimes b) \\
&= \min_{u \in \mathbb{R}_+^{p_1}, v \in \mathbb{R}_+^{p_2}} \min_{\substack{\Pi \in \mathbb{R}_+^{p_1 \times p_2} \\ \Pi \mathbf{1}_{p_2} = u, \ \Pi^T \mathbf{1}_{p_1} = v}} \sum_{i=1}^{p_1} \sum_{j=1}^{p_2} \Pi_{ij} C_{ij} + \rho D_{\mathrm{KL}}(u, a) + \rho D_{\mathrm{KL}}(v, b) + \varepsilon D_{\mathrm{KL}}(\Pi, a \otimes b).
\end{aligned}
$$

The inner minimization can be solved exactly by introducing dual variables $f \in \mathbb{R}^{p_1}, g \in \mathbb{R}^{p_2}$ and writing out the Lagrange dual problem

$$
\begin{aligned}
\min_{\substack{\Pi \in \mathbb{R}_+^{p_1 \times p_2} \\ \Pi \mathbf{1}_{p_2} = u, \Pi^T \mathbf{1}_{p_1} = v}} &\quad \sum_{i=1}^{p_1} \sum_{j=1}^{p_2} \Pi_{ij} C_{ij} + \varepsilon D_{\mathrm{KL}}(\Pi, a \otimes b) - f^T(\Pi \mathbf{1}_{p_2} - u) - g^T(\Pi^T \mathbf{1}_{p_1} - v) \\
&= \max_{f \in \mathbb{R}^{p_1}, \ g \in \mathbb{R}^{p_2}} \min_{\Pi \in \mathbb{R}_+^{p_1 \times p_2}} \sum_{i=1}^{p_1} \sum_{j=1}^{p_2} \Pi_{ij} C_{ij} + \varepsilon D_{\mathrm{KL}}(\Pi, a \otimes b) - f^T(\Pi \mathbf{1}_{p_2} - u) - g^T(\Pi^T \mathbf{1}_{p_1} - v).
\end{aligned}
$$

where we have removed the terms $\rho D_{\mathrm{KL}}(u, a)$ and $\rho D_{\mathrm{KL}}(v, b)$ since they do not depend on $\Pi$. Taking the gradient in $\Pi$ in the inner minimization and setting it to zero we get

$$C_{ij} + \varepsilon \log \left( \frac{\Pi_{ij}}{a_i b_j} \right) - f_i - g_j = 0$$

which implies that

$$\Pi_{ij} = a_i b_j \exp \left( \frac{f_i + g_j - C_{ij}}{\varepsilon} \right)$$

Now we can substitute this expression for $\Pi$ back into our Lagrange dual problem. First we compute

$$\varepsilon D_{\mathrm{KL}}(\Pi, a \otimes b) = \sum_{i=1}^{p_1} \sum_{j=1}^{p_2} \Pi_{ij} \left( f_i + g_j - C_{ij} \right) - \varepsilon \sum_{i=1}^{p_1} \sum_{j=1}^{p_2} \Pi_{ij} + \varepsilon \sum_{i=1}^{p_1} \sum_{j=1}^{p_2} a_i b_j$$

which implies that

$$
\begin{aligned}
&\sum_{i=1}^{p_1} \sum_{j=1}^{p_2} \Pi_{ij} C_{ij} + \varepsilon D_{\mathrm{KL}}(\Pi, a \otimes b) - f^T(\Pi \mathbf{1}_{p_2} - u) - g^T(\Pi^T \mathbf{1}_{p_1} - v) \\
&= u^T f + v^T g - \varepsilon \sum_{i=1}^{p_1} \sum_{j=1}^{p_2} a_i b_j \exp \left( \frac{f_i + g_j - C_{ij}}{\varepsilon} \right) + \varepsilon \sum_{i=1}^{p_1} \sum_{j=1}^{p_2} a_i b_j.
\end{aligned}
$$

Hence, the outer maximization in our Lagrange dual problem for $f$ and $g$ can now be written as

$$\max_{f \in \mathbb{R}^{p_1}, \ g \in \mathbb{R}^{p_2}} u^T f + v^T g - \varepsilon \sum_{i=1}^{p_1} \sum_{j=1}^{p_2} a_i b_j \exp \left( \frac{f_i + g_j - C_{ij}}{\varepsilon} \right)$$

where we have removed the last constant sum in $a_i b_j$. Finally we can rewrite our entire minimization from the start of this section as

$$\min_{\Pi \in \mathbb{R}_+^{p_1 \times p_2}} \mathcal{L}_{\rho,\varepsilon}(\Pi) = \min_{\substack{u \in \mathbb{R}_+^{p_1}, v \in \mathbb{R}_+^{p_2}}} \min_{\substack{\Pi \in \mathbb{R}_+^{p_1 \times p_2} \\ \Pi \mathbf{1}_{p_2} = u, \ \Pi^T \mathbf{1}_{p_1} = v}} \sum_{i=1}^{p_1} \sum_{j=1}^{p_2} \Pi_{ij} C_{ij} + \rho D_{\mathrm{KL}}(u, a) + \rho D_{\mathrm{KL}}(v, b) + \varepsilon D_{\mathrm{KL}}(\Pi, a \otimes b)$$

$$= \min_{u \in \mathbb{R}_+^{p_1}, v \in \mathbb{R}_+^{p_2}} \max_{f \in \mathbb{R}^{p_1}, g \in \mathbb{R}^{p_2}} u^T f + v^T g - \varepsilon \sum_{i=1}^{p_1} \sum_{j=1}^{p_2} a_i b_j \exp\left(\frac{f_i + g_j - C_{ij}}{\varepsilon}\right) + \rho D_{\mathrm{KL}}(u, a) + \rho D_{\mathrm{KL}}(v, b).$$

By strong duality, we can interchange the minimum and maximum above to write

$$\min_{\Pi \in \mathbb{R}_+^{p_1 \times p_2}} \mathcal{L}_{\rho,\varepsilon}(\Pi) = \max_{f \in \mathbb{R}^{p_1}, g \in \mathbb{R}^{p_2}} \min_{u \in \mathbb{R}_+^{p_1}, v \in \mathbb{R}_+^{p_2}} u^T f + v^T g - \varepsilon \sum_{i=1}^{p_1} \sum_{j=1}^{p_2} a_i b_j \exp\left(\frac{f_i + g_j - C_{ij}}{\varepsilon}\right) + \rho D_{\mathrm{KL}}(u, a) + \rho D_{\mathrm{KL}}(v, b)$$

$$= U^*(f) + V^*(g) - \varepsilon \sum_{i=1}^{p_1} \sum_{j=1}^{p_2} a_i b_j \exp\left(\frac{f_i + g_j - C_{ij}}{\varepsilon}\right)$$

where we define the functions

$$U^*(f) = \min_{u \in \mathbb{R}_+^{p_1}} u^T f + \rho D_{\mathrm{KL}}(u, a)$$
$$V^*(g) = \min_{v \in \mathbb{R}_+^{p_2}} v^T g + \rho D_{\mathrm{KL}}(v, b). \tag{35}$$

In fact, we can solve the minimizations in $U^*$ and $V^*$ in closed form to get the minimizers $u^* = a \odot \exp(-f/\rho)$ and $v^* = b \odot \exp(-g/\rho)$ which we can substitute back in to get

$$
\begin{aligned}
U^*(f) &= \sum_{i=1}^{p_1} u_i^* f_i + \rho \sum_{i=1}^{p_1} u_i^* \ln\left(\frac{u_i^*}{a_i}\right) - \rho \sum_{i=1}^{p_1} u_i^* + \rho \sum_{i=1}^{p_1} a_i \\
&= \sum_{i=1}^{p_1} u_i^* f_i - \sum_{i=1}^{p_1} u_i^* f_i - \rho \sum_{i=1}^{p_1} a_i \exp(-f_i/\rho) + \rho \sum_{i=1}^{p_1} a_i \\
&= -\rho \sum_{i=1}^{p_1} a_i \exp(-f_i/\rho) + \rho \sum_{i=1}^{p_1} a_i.
\end{aligned}
$$

Likewise we can see that

$$V^*(f) = -\rho \sum_{i=1}^{p_2} b_i \exp(-g_i/\rho) + \rho \sum_{i=1}^{p_2} b_i.$$

Thus, we can rewrite our full optimization as

$$\min_{\Pi \in \mathbb{R}_+^{p_1 \times p_2}} \mathcal{L}_{\rho,\varepsilon}(\Pi) = \max_{f \in \mathbb{R}^{p_1}, g \in \mathbb{R}^{p_2}} -\rho \sum_{i=1}^{p_1} a_i \exp(-f_i/\rho) - \rho \sum_{i=1}^{p_2} b_i \exp(-g_i/\rho) - \varepsilon \sum_{i=1}^{p_1} \sum_{j=1}^{p_2} a_i b_j \exp\left(\frac{f_i + g_j - C_{ij}}{\varepsilon}\right)$$

$$= \min_{f \in \mathbb{R}^{p_1}, g \in \mathbb{R}^{p_2}} \rho \sum_{i=1}^{p_1} a_i \exp(-f_i/\rho) + \rho \sum_{i=1}^{p_2} b_i \exp(-g_i/\rho) + \varepsilon \sum_{i=1}^{p_1} \sum_{j=1}^{p_2} a_i b_j \exp\left(\frac{f_i + g_j - C_{ij}}{\varepsilon}\right)$$

where we have removed the terms independent of $f$ and $g$.

Note that now we can optimize the cost function above by performing an alternating minimization on the dual variables $f$ and $g$. Taking the gradient in $f$ and setting it to zero we see that

$$-a_i \exp(-f_i/\rho) + a_i \sum_{j=1}^{p_2} b_j \exp\left(\frac{f_i + g_j - C_{ij}}{\varepsilon}\right) = 0$$

which implies that

$$f_i = -\frac{\varepsilon \rho}{\varepsilon + \rho} \ln\left(\sum_{j=1}^{p_2} b_j \exp\left(\frac{g_j - C_{ij}}{\varepsilon}\right)\right).$$

Similarly, we can write out

$$g_j = -\frac{\varepsilon \rho}{\varepsilon + \rho} \ln\left(\sum_{i=1}^{p_1} a_i \exp\left(\frac{f_i - C_{ij}}{\varepsilon}\right)\right).$$

We are now ready to write out the full unbalanced Sinkhorn algorithm which performs an alternating minimization on the dual potentials $f, g$ as outlined above. We remind the reader that the coupling matrix can be recovered from the dual potentials by the formula

$$\Pi_{ij} = a_i b_j \exp\left(\frac{f_i + g_j - C_{ij}}{\varepsilon}\right).$$

The unbalanced Sinkhorn algorithm proceeds as follows.

---

Algorithm 1. **UnbalancedSinkhorn**

---

**input**: Transport cost $C$, marginals $a, b$, marginal relaxation $\rho$, entropic regularization $\varepsilon$
**output**: Return the coupling matrix $\Pi$
Initialize $g = 0$
**while** $(f, g)$ has not converged **do**

  Set $f_i \leftarrow -\frac{\varepsilon\rho}{\varepsilon+\rho} \ln\left(\sum_{j=1}^{p_2} b_j \exp\left(\frac{g_j - C_{ij}}{\varepsilon}\right)\right)$ for $i \in [p_1]$

  Set $g_j \leftarrow -\frac{\varepsilon\rho}{\varepsilon+\rho} \ln\left(\sum_{i=1}^{p_1} a_i \exp\left(\frac{f_i - C_{ij}}{\varepsilon}\right)\right)$ for $j \in [p_2]$

Return the coupling matrix $\Pi_{ij} = a_i b_j \exp\left(\frac{f_i+g_j-C_{ij}}{\varepsilon}\right)$ for $i \in [p_1]$ and $j \in [p_2]$

---

The final output of the Sinkhorn algorithm optimization is a real-valued coupling matrix $\Pi \in \mathbb{R}_+^{p_1 \times p_2}$. In some cases, it is desirable to transform the coupling matrix into a binary-valued matching matrix $M \in \{0, 1\}^{p_1 \times p_2}$ with possibly an added restriction that there is at most one nonzero element in each row and column (to obtain a valid partial matching). This can be done by either thresholding the real matrix $\Pi$ or by assigning all maximal entries in each row (or column) to one and setting the remaining entries to zero. For our metabolomics matching problem, we describe our procedure for transforming our real-valued coupling into a binary matching matrix in the section on the GromovMatcher algorithm below.

## Gromov–Wasserstein

Now that we have introduced the general formulation of unbalanced optimal transport and its corresponding Sinkhorn algorithm, we can extend this formulation to matching problems between distributions of points that live in different dimensional spaces. In our metabolomics setting, we aim to match two datasets of $p_1$ and $p_2$ metabolic features respectively where each feature in a dataset is associated with a feature intensity vector $\{X_i\}_{i=1}^{p_1} \subset \mathbb{R}^{n_1}$ and $\{Y_j\}_{j=1}^{p_2} \subset \mathbb{R}^{n_2}$ respectively across samples. We assume that there exists a true matching matrix $M^* \in \{0, 1\}^{p_1 \times p_2}$ with at most one nonzero entry in each row and column such that two metabolites $(i, j)$ are matched if $M_{ij}^* = 1$.

We make the further assumption that if feature vectors $X_i, Y_j$ are matched and feature vectors $X_k, Y_l$ are matched under $M^*$, then we expect that

$$d_x(X_i, X_k) \approx d_y(Y_j, Y_l) \quad \text{if} \quad M_{ij}^* = M_{kl}^* = 1. \tag{36}$$

where $d_x$ is a distance metric on $\mathbb{R}^{n_1}$ and $d_y$ is a distance metric n $\mathbb{R}^{n_2}$. In practice, we always choose these distance metrics to be the normalized Euclidean distance defined for any $u, v \in \mathbb{R}^n$ as

$$d^{\text{euc}}(u, v) = \frac{1}{\sqrt{n}} \|u - v\| \tag{37}$$

which is equal to the cosine distance $d^{\cos}$ (i.e. one minus the correlation) for centered and scaled data. Given these two distance matrices $D^x = [d_x(X_i, X_k)]_{i,k=1}^{p_1} \in \mathbb{R}^{p_1 \times p_1}$ and $D^y = [d_y(Y_j, Y_l)]_{j,l=1}^{p_2} \in \mathbb{R}^{p_2 \times p_2}$ we would like to infer the true matching matrix $M^*$ by solving an optimization problem.

Consider the following objective function

$$\mathcal{E}(M) = \sum_{i,k=1}^{p_1} \sum_{j,l=1}^{p_2} M_{ij} M_{kl} \left| D_{ik}^x - D_{jl}^y \right|. \tag{38}$$

where the matching matrices $M \in \{0, 1\}^{p_1 \times p_2}$ we optimize over are constrained to satisfy marginal constraints $\Pi\mathbf{1}_{p_2} > 0$ and $\Pi^T\mathbf{1}_{p_1} > 0$. These marginal constraints simply impose that there is at least one nonzero entry in each row and column (i.e. each metabolite in both datasets has at least one corresponding match). Searching for the $\Pi$ minimizing $\mathcal{E}_{X,Y}(\Pi)$ consists of putting the non-zero entries in $\Pi$ such that the distance profiles of the matched features are similar, so that the minimizer of this criterion provides a good candidate estimate of $\Pi^*$. This is closely related to the Gromov–Hausdorff distance *Gromov, 2001*, an extension of optimal transport to the case where the sets to be coupled do not lie in the same metric space.

In practice, it is often desirable to optimize over a different set of matrices in order to make the optimization problem more tractable. Here we take intuition from optimal transport, and search over the set of coupling matrices with marginal constraints

$$\mathbf{U}(a, b) = \{\Pi \in \mathbb{R}_+^{p_1 \times p_2} : \Pi \mathbf{1}_{p_2} = a \text{ and } \Pi^T \mathbf{1}_{p_1} = b\}. \tag{39}$$

where as before, $a \in \mathbb{R}_+^{p_1}$ and $b \in \mathbb{R}_+^{p_2}$ are desired marginals which are typically set to be uniform distributions $a = \frac{1}{p_1}\mathbf{1}_{p_1}$ and $b = \frac{1}{p_2}\mathbf{1}_{p_2}$. These marginal vectors can be interpreted as distributions of masses $a_i$ and $b_j$ on the feature vectors $X_i$ and $Y_j$ respectively for $i \in [p_1], j \in [p_2]$.

Coupling matrices in $\mathbf{U}(a, b)$ transport the distribution of masses $a$ in the first dataset to the distribution of masses $b$ in the second dataset. Now we can formulate the Gromov–Wasserstein (GW) distance, introduced by **Mémoli, 2011**, as

$$\mathrm{GW}(a, b) = \min_{\Pi \in \mathbf{U}(a,b)} \mathcal{E}(\Pi) \tag{40}$$

By optimizing this objective, each entry $\Pi_{ij}$ now reflects the strength of the matched pair $(X_i, Y_j)$. Optimizing $\mathrm{GW}(a, b)$ then amounts to placing larger entries in $\Pi$ whose paired features have similar distance profiles. Before we develop an algorithm to optimize this objective, we first modify it to allow for unbalanced matchings where marginal constraints are not enforced exactly (e.g. features in both datasets can remain unmatched).

## Unbalanced Gromov–Wasserstein

In an untargeted context, all features measured in one study are not necessarily observed in another, either because these features are truly not shared or because of measurement error. However, the constraint $\Pi \in \mathbf{U}(a, b)$ in the original GW optimization criterion **Equation 40** ensures that all the mass is transported from one set to another, resulting in all features being matched across studies. In order to discard study-specific features during the GW computation, we use the unbalanced Gromov–Wasserstein (UGW) distance with an additional entropic regularization for computational purposes, described in **Sejourne et al., 2021**. The optimization problem therefore reads

$$\begin{aligned}\mathcal{L}_{\rho,\varepsilon}(\Pi) = \mathcal{E}(\Pi) \quad &+ \rho D_{\mathrm{KL}}\left(\Pi \mathbf{1}_{p_2} \otimes \Pi \mathbf{1}_{p_2}, a \otimes a\right) \\ &+ \rho D_{\mathrm{KL}}\left(\Pi^T \mathbf{1}_{p_1} \otimes \Pi^T \mathbf{1}_{p_1}, b \otimes b\right) \\ &+ \varepsilon D_{\mathrm{KL}}(\Pi \otimes \Pi, \left(a \otimes b\right)^{\otimes 2})\end{aligned} \tag{41}$$

$$\mathrm{UGW}_{\rho,\varepsilon} = \min_{\Pi \in \mathbb{R}_+^{p_1 \times p_2}} \mathcal{L}_{\rho,\varepsilon}(\Pi) \tag{42}$$

with $\rho, \varepsilon > 0$. Here $D_{\mathrm{KL}}$ is the Kullback–Leibler divergence defined in the previous sections and we define the tensor product $(P \otimes P)_{i,j,k,l} = P_{i,j}P_{k,l}$. Here we set the desired marginal constraints to $a = \frac{1}{p_1}\mathbf{1}_{p_1}$ and $b = \frac{1}{p_2}\mathbf{1}_{p_2}$ as before.

As in the case of unbalanced optimal transport (**Chizat et al., 2018**), the regularization $\rho$ times the Kullback–Leibler divergences allows for the relaxation of the marginal constraints $\Pi \mathbf{1}_{p_2} = a$ and $\Pi^T \mathbf{1}_{p_1} = b$. The value of $\rho > 0$ controls the extent to which we allow for mass destruction. Smaller values of $\rho$ tend to lessen the constraint on the marginals of $\Pi$, while balanced GW is recovered when $\rho \to +\infty$. As proposed in the original paper (**Sejourne et al., 2021**), our UGW cost modifies the UOT formulation by using the quadratic Kullback-Leibler divergence in $\Pi \mathbf{1}_{p_2} \otimes \Pi \mathbf{1}_{p_2}$ and $\Pi^T \mathbf{1}_{p_1} \otimes \Pi^T \mathbf{1}_{p_1}$ instead, hence preserving the quadratic form of the GW cost function $\mathcal{E}(\Pi)$.

The term $\varepsilon D_{KL}\left(\Pi \otimes \Pi, \left(a \otimes b\right)^{\otimes 2}\right)$ serves as an entropic regularization, inspired again by optimal transport. Adding such a penalty is a standard way to compute an approximate solution to the optimal transport problem using the Sinkhorn algorithm as we shall show in the following section. Here again, we modify the entropic penalty in UGW to have a quadratic form in $\Pi \otimes \Pi$ to agree with the quadratic form of the GW cost $\mathcal{E}(\Pi)$. The parameter $\varepsilon$ controls the smoothness (entropy) of the coupling matrix $\Pi$ where larger values of $\varepsilon$ encourage $\Pi$ to put uniform weights on many of its entries, leading to less precision in the feature matches. However, increasing $\varepsilon$ also leads to better numerical stability and a significant speedup of the alternating Sinkhorn algorithm used to optimize the objective function described below.

## UGW optimization algorithm

Now we are ready to write out an algorithm to optimize the UGW objective in *Equation 42*. First write our objective as

$$\mathcal{L}_{\rho,\varepsilon}(\Pi) = \sum_{i,k=1}^{p_1} \sum_{j,l=1}^{p_2} \Pi_{ij}\Pi_{kl}\left|D_{ik}^x - D_{jl}^y\right| \quad +\rho D_{\mathrm{KL}}\left(\Pi\mathbf{1}_{p_2} \otimes \Pi\mathbf{1}_{p_2}, a \otimes a\right)$$
$$+\rho D_{\mathrm{KL}}\left(\Pi^T\mathbf{1}_{p_1} \otimes \Pi^T\mathbf{1}_{p_1}, b \otimes b\right) \tag{43}$$
$$+\varepsilon D_{\mathrm{KL}}(\Pi \otimes \Pi, (a \otimes b)^{\otimes 2}).$$

Using the quadratic nature of our cost function, we aim to perform an alternating minimization in the two copies of $\Pi$. For the moment, let's differentiate these two copies by $\Pi$ and $\Gamma$ and write the new cost

$$\mathcal{F}_{\rho,\varepsilon}(\Pi,\Gamma) = \sum_{i,k=1}^{p_1} \sum_{j,l=1}^{p_2} \Pi_{ij}\Gamma_{kl}\left|D_{ik}^x - D_{jl}^y\right| \quad +\rho D_{\mathrm{KL}}(\Pi\mathbf{1}_{p_2} \otimes \Gamma\mathbf{1}_{p_2}, a \otimes a)$$
$$+\rho D_{\mathrm{KL}}(\Pi^T\mathbf{1}_{p_1} \otimes \Gamma^T\mathbf{1}_{p_1}, b \otimes b) \tag{44}$$
$$+\varepsilon D_{\mathrm{KL}}(\Pi \otimes \Gamma, (a \otimes b)^{\otimes 2}).$$

Before we expand this cost, we introduce the notation $m(\pi)$ to denote the sum of the elements of $\pi$ which can be a vector, matrix or tensor. In general, for four positive distributions $\pi, a \in \mathbb{R}_+^p$ and $\gamma, b \in \mathbb{R}_+^q$ we have that the KL satisfies the tensorization property

$$D_{\mathrm{KL}}(\pi \otimes \gamma, a \otimes b) = \sum_{i=1}^p \sum_{j=1}^q \pi_i\gamma_j \ln\left(\frac{\pi_i\gamma_j}{a_ib_j}\right) - \sum_{i=1}^p \sum_{j=1}^q \pi_i\gamma_j + \sum_{i=1}^p \sum_{j=1}^q a_ib_j$$
$$= m(\gamma)\sum_{i=1}^p \pi_i \ln\left(\frac{\pi_i}{a_i}\right) + m(\pi)\sum_{j=1}^q \gamma_j \ln\left(\frac{\gamma_j}{b_j}\right) - m(\pi)m(\gamma) + m(a)m(b) \tag{45}$$
$$= m(\gamma)D_{\mathrm{KL}}(\pi,a) + m(\pi)D_{\mathrm{KL}}(\gamma,b) + \left(m(\pi) - m(a)\right)\left(m(\gamma) - m(b)\right).$$

Specifically, if we remove those terms that do not depend on $\gamma$ we are left with

$$D_{\mathrm{KL}}(\pi \otimes \gamma, a \otimes b) = m(\pi)D_{\mathrm{KL}}(\gamma,b) + m(\gamma)\sum_{i=1}^p \pi_i \ln\left(\frac{\pi_i}{a_i}\right) + const. \tag{46}$$

This allows us to write for the marginal constraints $a \in \mathbb{R}_+^{p_1}, b \in \mathbb{R}_+^{p_2}$ and couplings $\Pi, \Gamma \in \mathbb{R}_+^{p_1 \times p_2}$ that

$$D_{\mathrm{KL}}(\Pi\mathbf{1}_{p_2} \otimes \Gamma\mathbf{1}_{p_2}, a \otimes a) = m(\Pi)D_{\mathrm{KL}}(\Gamma\mathbf{1}_{p_2}, a) + m(\Gamma)\sum_{i=1}^{p_1}(\Pi\mathbf{1}_{p_2})_i \ln\left(\frac{(\Pi\mathbf{1}_{p_2})_i}{a_i}\right) + const.$$
$$D_{\mathrm{KL}}(\Pi^T\mathbf{1}_{p_1} \otimes \Gamma^T\mathbf{1}_{p_1}, b \otimes b) = m(\Pi)D_{\mathrm{KL}}(\Gamma^T\mathbf{1}_{p_1}, b) + m(\Gamma)\sum_{j=1}^{p_2}(\Pi^T\mathbf{1}_{p_1})_j \ln\left(\frac{(\Pi^T\mathbf{1}_{p_1})_j}{b_j}\right) + const.$$
$$D_{\mathrm{KL}}(\Pi \otimes \Gamma, (a \otimes b)^{\otimes 2}) = m(\Pi)D_{\mathrm{KL}}(\Gamma, a \otimes b) + m(\Gamma)\sum_{i=1}^{p_1}\sum_{j=1}^{p_2} \Pi_{ij} \ln\left(\frac{\Pi_{ij}}{a_ib_j}\right) + const.$$

where in the expansions above we have removed all terms that are independent of $\Gamma$. Finally, expanding out $\mathcal{F}_{\rho,\varepsilon}(\Pi,\Gamma)$ and keeping only those terms that depend on $\Gamma$ we get

$$\mathcal{F}_{\rho,\varepsilon}(\Pi,\Gamma) = \sum_{k=1}^{p_1}\sum_{l=1}^{p_2} \Gamma_{kl}C_{kl}^{\Pi} + \rho m(\Pi)D_{\mathrm{KL}}(\Gamma\mathbf{1}_{p_2}, a) + \rho m(\Pi)D_{\mathrm{KL}}(\Gamma^T\mathbf{1}_{p_1}, b) + \varepsilon m(\Pi)D_{\mathrm{KL}}(\Gamma, a \otimes b) \tag{47}$$

where the cost matrix $C^{\Pi} \in \mathbb{R}^{p_1 \times p_2}$ is defined as

$$C_{kl}^{\Pi} = \sum_{i=1}^{p_1}\sum_{j=1}^{p_2} \Pi_{ij}\left|D_{ik}^x - D_{jl}^y\right| + \rho\sum_{i=1}^{p_1}(\Pi\mathbf{1}_{p_2})_i \ln\left(\frac{(\Pi\mathbf{1}_{p_2})_i}{a_i}\right) + \rho\sum_{j=1}^{p_2}(\Pi^T\mathbf{1}_{p_1})_j \ln\left(\frac{(\Pi^T\mathbf{1}_{p_1})_j}{b_j}\right) + \varepsilon\sum_{i=1}^{p_1}\sum_{j=1}^{p_2} \Pi_{ij} \ln\left(\frac{\Pi_{ij}}{a_ib_j}\right). \tag{48}$$

where we have hidden the dependence of $C^{\Pi}$ on the distance matrices $D^x, D^y$, the marginals $a, b$, and the regularization parameters $\rho, \varepsilon$ for ease of notation.

Remarkably, the cost above in $\Gamma$ for fixed $\Pi$ is in the form of an unbalanced optimal transport problem which can be solved through unbalanced Sinkhorn iterations (Algorithm 1). Note that in our

derivation above, it did not matter whether we optimized $\Gamma$ with $\Pi$ fixed or vice versa because the cost $\mathcal{F}_{\rho,\varepsilon}(\Pi, \Gamma)$ is symmetric in both of its arguments.

Our iterative algorithm for solving the unbalanced GW problem will proceed at each iteration by optimizing $\Gamma$ to minimize the cost above using the unbalanced Sinkhorn method, setting $\Pi$ equal to $\Gamma$ and repeating. With each iteration, we expect this iterative procedure to make smaller and smaller updates to $\Gamma$ until convergence. By definition, at the end of each iteration we assign $\Pi = \Gamma$ so the minimizer of $\mathcal{F}_{\rho,\varepsilon}(\Pi, \Gamma)$ we converge to should also be a minimizer of the original UGW cost $\mathcal{L}_{\rho,\varepsilon}(\Pi)$ in the sense that the relaxation of $\mathcal{L}_{\rho,\varepsilon}(\Pi)$ to $\mathcal{F}_{\rho,\varepsilon}(\Pi, \Gamma)$ is tight. This is proven rigorously under strict mathematical assumptions in *Sejourne et al., 2021*. We state the full UGW optimization algorithm below.

---

Algorithm 2. **UnbalancedGromovWasserstein**

---

**input**: Distance matrices $D^x, D^y$, marginals a, b marginal relaxation $\rho$, entropic regularization $\varepsilon$
**output**: Return the coupling matrix $\Pi$
Initialize $\Pi = \Gamma = a \otimes b / \sqrt{m(a)m(b)}$
**while** $(\Pi, \Gamma)$ has not converged **do**
 Update $\Pi \leftarrow \Gamma$
 Update $\Gamma = \text{UnbalancedSinkhorn}(C^{\Pi}, a, b, \rho m(\Pi), \varepsilon m(\Pi))$
 Rescale $\Gamma \leftarrow \sqrt{m(\Pi)/m(\Gamma)}\Gamma$
Update $\Pi \leftarrow \Gamma$
Return $\Pi$.

---

Following the implementation of the UGW algorithm in *Sejourne et al., 2021*, we initialize both $\Pi$ and $\Gamma$ to be the product distribution of the marginals $a \otimes b / \sqrt{m(a)m(b)}$ before we begin the optimization. Also, we note that if $(\Pi, \Gamma)$ is a minimizer of our UGW objective $\mathcal{F}_{\rho,\varepsilon}(\Pi, \Gamma)$, then so is $(\frac{1}{s}\Pi, s\Gamma)$ for any scale factor $s > 0$. Hence, we can set $m(\frac{1}{s}\Pi) = m(s\Gamma)$ by choosing $s = \sqrt{m(\Pi)/m(\Gamma)}$. This motivates the final step in the while loop of the UGW algorithm where the rescaling of $\Gamma$ by the factor $\sqrt{m(\Pi)/m(\Gamma)}$ leads to mass equality $m(\Pi) = m(\Gamma)$ and also stabilizes the convergence of the algorithm.

Returning to our metabolomics matching problem, we further guide our UGW optimization procedure by discouraging it from matching metabolic feature pairs whose mass-to-charge ratios are incompatible. Namely, we choose a value $m_{\text{gap}}$ such that for all pairs $(i, j)$ with $i \in [p_1], j \in [p_2]$ and mass-to-charge ratios $m_i^x, m_j^y$ we enforce that

$$|m_i^x - m_j^y| > m_{\text{gap}} \implies \Pi_{ij} = 0. \tag{49}$$

In practice, this is done by taking the optimal transport cost $C^{\Pi}$ in every iteration of the UGW algorithm and premultiplying it elementwise by a factor $W \in \mathbb{R}_+^{p_1 \times p_2}$ given by

$$C^{\Pi} \to W \odot C^{\Pi}, \quad W_{ij} = 99 \cdot \mathbf{1}_{\{|m_i^x - m_j^y| > m_{\text{gap}}\}} + 1 \tag{50}$$

where $\mathbf{1}_{\mathcal{X}}$ denotes the indicator function that is one when the condition $\mathcal{X}$ is satisfied and zero otherwise. Such a prefactor changes the transport cost to be very large for feature matches with incompatible mass-to-charge ratio times, and hence, the entries of $\Pi$ set small weights at these entries. Our weighted UGW algorithm is rewritten below.

---

Algorithm 3. **WeightedUnbalancedGromovWasserstein**

---

**input** : Distance matrices $D^x, D^y$, marginals $a, b$, marginal relaxation $\rho$, entropic regularization $\varepsilon$,
 mass-to-charge ratios $m^x, m^y$, mass-to-charge ratio gap $m_{\text{gap}}$
**output**: Return the coupling matrix $\Pi$
Initialize $\Pi = \Gamma = a \otimes b / \sqrt{m(a)m(b)}$
Set $W_{ij} = 99 \cdot \mathbf{1}_{\{|m_i^x - m_j^y| > m_{\text{gap}}\}} + 1$ for $i \in [p_1]$ and $j \in [p_2]$
**while** $(\Pi, \Gamma)$ has not converged **do**
 Update $\Pi \leftarrow \Gamma$
 Update $\Gamma = \text{UnbalancedSinkhorn}(W \odot C^{\Pi}, a, b, \rho m(\Pi), \varepsilon m(\Pi))$
 Rescale $\Gamma \leftarrow \sqrt{m(\Pi)/m(\Gamma)}\Gamma$
Update $\Pi \leftarrow \Gamma$
Return $\Pi$.

---

As mentioned before, the coupling matrix returned by our weighted UGW algorithm is a real-valued matrix rather than a binary matching matrix. In the next section, we describe how we incorporate metabolite retention time information to filter out unlikely pairs in our coupling matrix and transform it into a valid one-to-one matching of features across two datasets.

### Retention time drift estimation and filtering

To filter out unlikely matches from the coupling matrix returned by Algorithm 3 above, we use the retention times (RTs) of the metabolites in both datasets. We remind the reader that RTs were not incorporated into the weighted UGW algorithm since they often exhibit a non-linear deviation between datasets, and hence are not directly comparable. However, using the metabolite coupling $\tilde{\Pi} \in \mathbb{R}_+^{p_1 \times p_2}$ obtained from Algorithm 3, it is possible to estimate this RT drift. The estimated RT drift $\hat{f}: \mathbb{R}_+ \to \mathbb{R}_+$ allows us to assess the plausibility of the pairs recovered by the restricted UGW coupling $\tilde{\Pi}$, and discard pairs incompatible with the estimated drift.

We propose to learn the drift $\hat{f}$ through the weighted spline regression

$$\min_{f \in \mathcal{B}_{n,k}} \sum_{i=1}^{p_1} \sum_{j=1}^{p_2} \tilde{\Pi}_{ij} \left| f\left(RT_i^x\right) - RT_j^y \right| \tag{51}$$

where $\mathcal{B}_{n,k}$ is the set of $n$-order B-splines with $k$ knots. All pairs $(RT_i^x, RT_j^y)$ in objective *Equation 51* are weighted by the coefficients of $\tilde{\Pi}$ so that larger weights are given to pairs identified with high confidence in the first step of our procedure.

Pairs identified as incompatible with the estimated RT drift are then discarded from the coupling matrix. To do this, we first take the estimated RT drift $\hat{f}$, and the set of pairs $\mathcal{S} = \{i, j : \tilde{\Pi}_{i,j} \neq 0\}$ recovered in $\tilde{\Pi}$ with nonzero entries. We then define the residual associated with $(i, j) \in \mathcal{S}$ as

$$r_{\hat{f}}(i,j) = |\hat{f}(RT_i^x) - RT_j^y|. \tag{52}$$

The 95% prediction interval and the median absolute deviation (MAD) of these residuals are given by

$$\mathrm{PI} = 1.96 \times \mathrm{std}(\{r_{\hat{f}}(i,j), (i,j) \in \mathcal{S}\})$$

$$\mathrm{MAD} = \mathrm{median}(\{|r_{\hat{f}}(i,j) - \mu_r|, (i,j) \in \mathcal{S}\}) \tag{53}$$

$$\mu_r = \mathrm{median}(\{|r_{\hat{f}}(i,j)|, (i,j) \in \mathcal{S}\})$$

where $|\mathcal{S}|$ is the size of $\mathcal{S}$ and the functions std, median denote the standard deviation and median respectively. Following *Habra et al., 2021*, we then create a new filtered coupling matrix $\widehat{\Pi} \in \mathbb{R}_+^{p_1 \times p_1}$ given by

$$\widehat{\Pi}_{ij} = \begin{cases} \tilde{\Pi}_{ij} & \text{if } r_{\hat{f}}(i,j) < \mu_r + r_{\text{thresh}} \\ 0 & \text{otherwise} \end{cases}. \tag{54}$$

where $r_{\text{thresh}}$ is a given filtering threshold. The procedure of estimating the drift function $\hat{f}$ in *Equation 51* and filtering the coupling can be repeated for multiple iterations, to improve the drift and coupling estimation. In our main algorithm, we use two preliminary iterations where we estimate the RT drift and discard outliers with $r_{\text{thresh}} = \mathrm{PI}$, defined as points falling outside of the 95% prediction interval. We the re-estimate the drift and perform a final filtering step with the more stringent MAD by setting $r_{\text{thresh}} = 2 \times \mathrm{MAD}$.

At this stage, it is possible for $\widehat{\Pi}$ to still contain coefficients of very small magnitude. As an optional postprocessing step, we discard these coefficients by setting all entries smaller than $\tau \max(\widehat{\Pi})$ to zero for some scaling constant $\tau \in [0, 1]$. Lastly, a feature from either study could have multiple possible matches, since $\widehat{\Pi}$ can have more than one non-zero coefficient per row or column. Although reporting multiple matches can be helpful in an exploratory context, for the sake of simplicity in our analysis, the final output of GromovMatcher returns a one-to-one matching. Consequently, we only keep those metabolite pairs $(i, j)$ where the entry $\widehat{\Pi}_{ij}$ is largest in its corresponding row and column. All nonzero entries of $\widehat{\Pi}$ which do not satisfy this criterion are set to zero. Finally, we convert $\widehat{\Pi}$ into a

binary matching matrix $M \in \{0,1\}^{p_1 \times p_2}$ with ones in place of its nonzero entries and this final output is returned to the user.

As a naming convention, we use the abbreviation GM for our GromovMatcher method, and use the abbreviation GMT when running GromovMatcher with the optional $\tau$-thresholding step.

## GromovMatcher algorithm summary

In summary, our full GromovMatcher algorithm consists of (1) UGW optimization followed by (2) retention time drift estimation and filtering.

The tuning of $\rho$ and $\epsilon$ was computationally driven and the two parameters were set as low as possible, with $\rho = 0.05$ and $\epsilon = 0.005$. Based on literature (*Loftfield et al., 2021*; *Hsu et al., 2019*; *Climaco Pinto et al., 2022*; *Habra et al., 2021*; *Chen et al., 2021*) and what is considered to be a plausible variation of a feature's $m/z$, we set $m_{\text{gap}} = 0.01$ ppm. For RT drift estimation, the order of the B-splines was set to $n = 3$ by default, while the number of knots $k$ was selected by 10-fold cross-validation. If the optional thresholding step was applied in GMT, we set $\tau = 0.3$. Otherwise, we let $\tau = 0$ which gives the unthresholded GM algorithm.

---

Algorithm 4. **GromovMatcher**

---

**input** : Distance matrices $D^x, D^y$, marginals $a, b$, marginal relaxation $\rho$, entropic regularization $\varepsilon$,
 mass-to-charge ratios $m^x, m^y$, mass-to-charge ratio gap $m_{\text{gap}}$,
 retention times $RT^x, RT^y$, B-spline order $n$, filtering threshold $\tau$
**output**: Return the matching matrix $M$ and the retention time drift $\hat{f}$
# Step 1: Weighted UGW optimization
Compute $\tilde{\Pi} = \text{WeightedUnbalancedGromovWasserstein}(D^x, D^y, a, b, \rho, \varepsilon, m^x, m^y)$
# Step 2: Retention time drift estimation and filtering
for $i = 1 : 3$ do

 Perform weighted spline regression **Equation 51** for RT drift $\hat{f} \in \mathcal{B}_{n,k}$ where $k$ is chosen by 10-fold cross validation
 Initialize $r_{\text{thresh}} = 0$
 if $i < 3$ then
 Set $r_{\text{thresh}} = \text{PI}$ from **Equation 53**
 else
 Set $r_{\text{thresh}} = 2 \times \text{MAD}$ from **Equation 53**
 Set $\tilde{\Pi} = \widehat{\Pi}$

Compute $\mathcal{U} = \max(\widehat{\Pi})$
Set $\widehat{\Pi}_{ij} = 0$ if $\widehat{\Pi}_{ij} < \tau\mathcal{U}$ for $i \in [p_1]$ and $j \in [p_2]$
Set $\widehat{\Pi}_{ij} = 0$ if $i \neq \text{argmax}_k \widehat{\Pi}_{k,j}$ or $j \neq \text{argmax}_k \widehat{\Pi}_{i,k}$ for $i \in [p_1]$ and $j \in [p_2]$
Define the binarized matching $M_{ij} = \mathbf{1}_{\{\widehat{\Pi}_{ij} > 0\}}$
Return $M$ and $\hat{f}$.

---

# Appendix 2

Here we discuss existing metabolomic alignments methods and the hyperparameter experiments we perform on these methods. We consider two existing alignment methods for comparison, metabCombiner (*Habra et al., 2021*) and M2S (*Climaco Pinto et al., 2022*). Both of them take the same kind of input as GromovMatcher, i.e. feature tables with features identified with their *m/z*, RT, and intensities across samples.

## MetabCombiner hyperparameter experiments

MetabCombiner (*Habra et al., 2021*) is a three-step process that begins by grouping features based on their *m/z* within user-specified bins. This creates a search space for potential feature pairs. In the second step, MetabCombiner estimates the RT drift using the potential feature pairs identified in the first step, and eliminates outlying pairs over several iterations. This step can incorporate prior knowledge by identifying shared features and marking them as anchors, which are not discarded. In the final step, MetabCombiner scores the remaining feature pairs based on their *m/z*, RT, and relative intensity compatibility to discriminate between multiple matches for one feature. The scoring system relies on weights assigned to *m/z*, RT, and feature intensities, with the magnitude of those weights reflecting the reliability of the corresponding measurements across studies.

MetabCombiner (*Habra et al., 2021*) includes adjustable parameters throughout the pipeline. We set most of them to default values unless otherwise stated. MetabCombiner first establishes candidate pairs by binning features in the *m/z* dimension with a width of binGap, and pairing the features sorted by relative intensities. The 'binGap' parameter sets the *m/z* tolerance of metabCombiner, similar to $m_{gap}$ in GromovMatcher. We used the same value of 0.01 as in GromovMatcher.

MetabCombiner then estimates the RT drift using basis splines, and removes pairs associated with a high residual (twice the mean model error) from the candidate set.

In our main experiment, the RT drift is estimated exclusively using candidate pairs selected by the pipeline. However, it is also possible to include known ground truth pairs as 'anchors' to estimate the RT drift. We choose not to rely on prior knowledge for drift estimation as *Habra et al., 2021* show their drift estimation to be efficient and robust, even without prior knowledge. To confirm this claim, we conduct a sensitivity analysis comparing the results obtained in our main experiment with those obtained when supplying metabCombiner with known shared metabolites to anchor the RT drift estimation. We randomly select 100 anchors from the ground truth matching and compute the metabCombiner matchings with otherwise identical settings as in our main experiment. The results from this analysis (reported in *Appendix 2—figure 1*) show that the unsupervised RT drift estimation (using anchors selected by the pipeline only) performs as well as the supervised RT drift estimation, showing the drift estimation to be very consistent, with or without shared entities.

After establishing candidate pairs and filtering out those that contradict the estimated RT drift, metabCombiner discriminates between multiple matches using a scoring system that considers *m/z*, RT, and rankings of the median feature intensities. Each dimension has a specific weight that can be left at default, manually adjusted, or automatically tuned using known matched pairs. *Habra et al., 2021* provide qualitative guidelines for tuning the weights manually, mainly based on the experimental conditions and visual inspection of the RT drift plot. Since this approach is difficult to implement in the various settings we consider for our simulation study, we rely on the quantitative tuning function included in the metabCombiner pipeline. This function takes into account known shared features and tunes the weights to optimize the scores of those known matches. We randomly select 100 known true matches to define the objective function metabCombiner maximizes. We search over the recommended range of values, with the *m/z* weight $A \in [50, 150]$, the RT weight $B \in [5, 20]$ and the feature intensities weight $C \in [0, 1]$. *Appendix 2—figure 1* presents the results obtained with the weights set at default values ($A = 100, B = 15, C = 0.5$), as a sensitivity analysis.

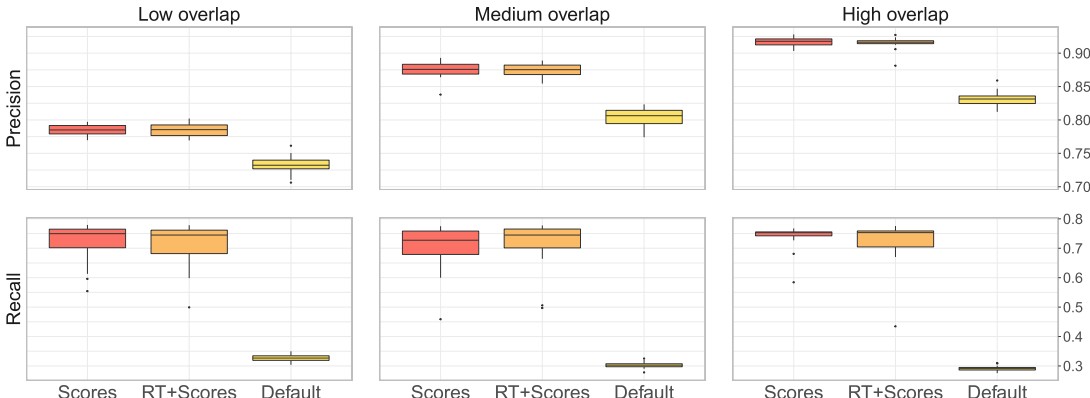

**Appendix 2—figure 1.** Performance of metabCombiner with the different parameter settings. The first setting, labelled 'Scores' correspond to the design of our main analysis, where 100 randomly selected true pairs are supplied to metabCombiner to set the scoring weights automatically, but are not otherwise used. In the second setting, labelled 'Scores + RT', metabCombiner is allowed to use the 100 true pairs not only to set the scoring weights, but also to estimate the RT drift. Finally, in the third 'Default' setting, we do not use any prior knowledge for the RT drift estimation and keep the scoring weights' default values.

## M2S hyperparameter experiments

*Climaco Pinto et al., 2022* introduce M2S as a more versatile alternative to metabCombiner, while still adhering to most of its core principles. Like metabCombiner, M2S follows a three-step process. First it searches for matches within user-defined thresholds for $m/z$, retention time, and mean feature intensity. Next, M2S estimates $m/z$, RT and feature intensity drifts between datasets and removes any outlier pairs. Finally, M2S selects the best match using a scoring system that weighs each measurement, similar to metabCombiner. M2S notably stands out by providing greater flexibility in the methods and measurements used at each step of the procedure, resulting however in a larger number of parameters that require manual fine-tuning. To address this, we adopt two different approaches for the simulation study and the EPIC study alignment. In the simulation study, we set the initial thresholds to oracle values and investigate technical parameters. For the EPIC study alignment, we use the combination of technical parameters with the best average F1-score in the simulation study and select the best threshold values based on the performance on the validation subset.

More precisely, M2S first matches all pairs of metabolic features whose absolute difference in $m/z$, RT, and median of $log_{10}$ FI are within the user-defined thresholds 'MZ_intercept', 'RT_intercept' and 'log10FI_intercept'. On simulated data experiments, we set these thresholds to MZ_intercept = 0.01, RT_intercept = 3.5 and log10FI_intercept = 0.2 which are large enough to not exclude any true feature matches in any of the scenarios for our simulated data under low, medium, and high overlap/noise (see Methods). M2S also offers more detailed options to match features whose absolute difference stays within two lower and upper bound lines with a given slope where the intercepts of these lines are defined using the values above. In our analysis, we set the slopes of these linear boundaries to zero so as to not remove any true matches. Because the reference and target studies we are matching in the simulated analysis are on the same scale, we set the FI adjust method to 'none'.

The second step of M2S involves calculating penalization scores for every pair of matches which are used to determine the best set of matches between metabolic features of both datasets. This step depends on a set of hyperparameters which we perform a grid search over to optimize the performance of M2S. For estimating the $m/z$, RT, and FI drift, the hyperparameters are the percentage of neighbors 'nrNeighbors', the neighborhood shape 'neighMethod', and the LOESS span percentage 'pctPointsLoess' used to smooth the estimated drift functions. After the drifts are estimated, they are normalized using a method specified by 'residPercentile' that puts the $m/z$, RT, and FI residuals on the same scale. We always fix residPercentile = NaN which defaults to the standard 2 × MAD normalization. Next, for every remaining metabolic feature match, the residuals/ drifts of the $m/z$, RT, and FI are added together by taking the weighted square root sum of squares. For unnormalized data where feature intensity magnitudes are important, we weight all three drifts

equally using $W = (1, 1, 1)$ and for data with normalized feature intensities we set the FI drift weight to zero such that $W = (1, 1, 0)$. Finally, using these weighted penalization scores, M2S selects the best matched pair within a multiple match cluster to obtain a one-to-one matching between datasets.

The third and final step of M2S involves removing those remaining matches which have large differences in *m/z*, RT, or FI. This can be performed using several methods indicated by the hyperparameter 'methodType'. Each method excludes those matched pairs whose differences in *m/z*, RT, or FI exceed a certain number of median absolute deviations indicated by the parameter 'nrMAD'. The remaining one-to-one metabolic feature matches are returned as the final result of the M2S algorithm.

To optimally tune M2S on our simulated experiments, we determine the optimal M2S parameter combination for each individual simulation setting (low, medium, high overlap and noise) by performing a grid search over the product of parameter lists

- nrNeighbors = [0.01, 0.05, 0.1, 0.5, 1]
- neighMethod = ['cross', 'circle']
- pctPointsLoess = [0, 0.1, 0.5]
- methodType = ['none', 'scores', 'byBins', 'trend_mad', 'residuals_mad']
- nrMAD = [1, 3, 5]

Each parameter combination for M2S is tested across 20 randomly generated datasets at the same overlap and noise settings. For each setting, the combination of parameters above with the best average F1-score across these 20 trials is used as the optimal parameter choice.

M2S applies initial RT thresholds to search for candidate pairs, which may favor settings where the RT drift follows a linear trend. Therefore, as a sensitivity analysis, we apply M2S to simulated data with a linear drift. The simulation process is identical to that of our main simulation study, except for the deviation of the RT in dataset 2. Specifically, for a given overlap value, we divide the original real-world dataset into two smaller datasets and introduce random noise to the *m/z*, RT and intensities of the features, without introducing a systematic deviation to the RT in dataset 2. M2S parameters are kept identical to the ones used in our main analysis in comparable settings. The results obtained by M2S on three pairs of datasets generated for three overlap values (0.25, 0.5 and 0.75) and a medium noise level are reported in *Appendix 2—table 1*. While the results obtained in a high overlap setting are close to those obtained in our main analysis M2S demonstrates better performance in a low overlap setting when the RT drift is linear than in our main analysis. This observation is consistent with the results obtained by M2S on EPIC data, considering the relatively low estimated overlap between the aligned EPIC studies in our main analysis.

**Appendix 2—table 1.** Performance of M2S in a setting where the RT drift between studies is linear.

| Metric | Low overlap | Medium overlap | High overlap |
|---|---|---|---|
| Precision | 0.831 | 0.917 | 0.947 |
| Recall | 0.934 | 0.933 | 0.939 |

For the EPIC data, we select the parameter combination that yields the highest F1-score across all simulated settings. However, due to the unavailability of oracle values for setting initial thresholds, we perform a search over several MZ intercept values (0.01, 0.05, and 0.1), RT intercept values (0.1, 0.5, 1, and 5), and logFI intercept values (1, 10, and 100).

## Appendix 3

In this section, we study the sensitivity of all three alignment methods GMT, M2S, and mC to the validation dataset split when creating two validation studies for matching. As described in the section "Validation on ground-truth data" and depicted in *Figure 2* of the main text, we generate two datasets to be matched by splitting an initial LC-MS dataset with $p$ features and $n$ samples into two smaller overlapping datasets. The first dataset has $p_1$ features and $n_1$ samples while the second dataset has $p_2$ features and $n_2$ samples. The sets of samples in both datasets are disjoint such that $n_1 + n_2 = n$. However, the dataset split is constructed such that both datasets share $\approx \lambda p$ of their features where $\lambda \in [0, 1]$ is an overlap fraction. Namely, this is done by defining the dataset feature sizes as

$$p_1 = \left\lfloor \left(\lambda + \lambda_f(1 - \lambda)\right)p \right\rfloor, \quad p_2 = \left\lfloor \left(\lambda + (1 - \lambda_f)(1 - \lambda)\right)p \right\rfloor \tag{55}$$

and the dataset sample sizes as

$$n_1 = \lfloor \lambda_s n \rfloor, \quad n_2 = n - \lfloor \lambda_s n \rfloor. \tag{56}$$

As before, $\lfloor \cdot \rfloor$ and $\lceil \cdot \rceil$ denote integer floor and ceiling functions. Then taking the original LC-MS dataset and randomly permuting its samples and features, the first $p_1$ features and first $n_1$ samples are placed into dataset 1 while the last $p_2$ features and last $n_2$ samples are placed into dataset 2. It is indeed easy to check here that with such a splitting procedure, the feature overlap between both datasets is $p_1 + p_2 - p \approx \lambda p$.

Here $\lambda_f \in [0.5, 1]$ controls the fraction of features in dataset 1 that is not shared with dataset 2 and $\lambda_s \in (0, 1)$ controls the fraction of samples in dataset 1 vs. dataset 2. In particular, if $\lambda_f = 1$ then the features in dataset 2 are entirely a subset of those in dataset 1. In the experiments described in the main text, we always set $\lambda_f = \lambda_s = 0.5$ as to balance the number of features and samples in both resulting datasets.

Now we study how the performance of all three alignment methods changes when $\lambda_f$, $\lambda_s$ and the feature overlap $\lambda$ are varied. Here we vary the feature overlap $\lambda \in \{0.25, 0.5, 0.75\}$, the feature fraction $\lambda_f \in \{0.5, 0.6, 0.7, 0.8, 0.9\}$, and the sample fraction $\lambda_s \in \{0.1, 0.2, \ldots, 0.9\}$. In *Appendix 3—figures 1–3* we show how the precision and recall of GMT, M2S, and mC depend on these parameters. Here we use the same unnormalized validation data and experimental setup as decribed in the main text section "Validation on ground-truth data" and in the Methods and Materials section "Validation on simulated data". For each triple $(\lambda, \lambda_f, \lambda_s)$ we randomly generate 20 dataset splits with these parameters and show the average precision and recall for each method over these trials. Our method GMT (thresholded GromovMatcher) is applied out-of-box with the default hyperparameter settings. The algorithm hyperparameters for mC and M2S are chosen optimally for each individual triple of dataset parameters $(\lambda, \lambda_f, \lambda_s)$ to maximize the average F1 score in each setting. The hyperparameters searched over when optimizing mC and M2S are described in detail in Appendix 2.

Consistent with prior validation experiments, we find that GromovMatcher outperforms both mC and M2S in all dataset regimes, for low overlap and high overlap $\lambda$ as well as for varying balances of features $\lambda_f$ and samples $\lambda_s$. Remarkably, all three methods exhibit the same sensitivity to variations of $(\lambda, \lambda_f, \lambda_s)$. All methods exhibit a monotonic decrease in their precision as $\lambda_f$ drops from 0.9 to 0.5. In other words, the most challenging setting for matching both datasets is when dataset 1 and dataset 2 both have an equal number of unshared features (e.g. $\lambda_f = 0.5$). Likewise, the simplest setting for matching is when the features in dataset 2 are exactly a subset of the features in dataset 1 (e.g. $\lambda_f = 1$). Sensitivity to this parameter $\lambda_f$ is most noticeable at low feature overlap $\lambda = 0.25$.

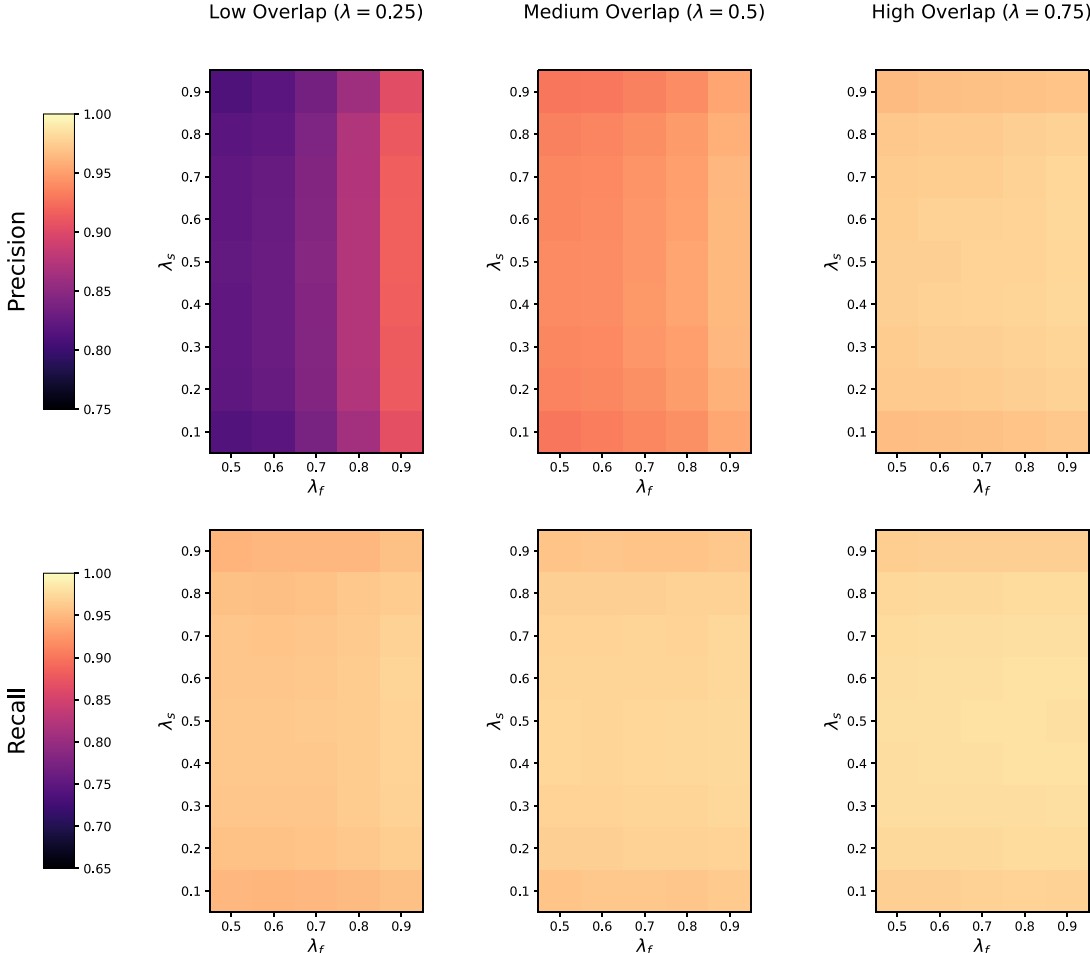

**Appendix 3—figure 1.** Sensitivity of thresholded GromovMatcher (GMT) to feature overlap fraction $\lambda$, feature imbalance fraction $\lambda_f$, and sample imbalance fraction $\lambda_s$ between two datasets being matched.

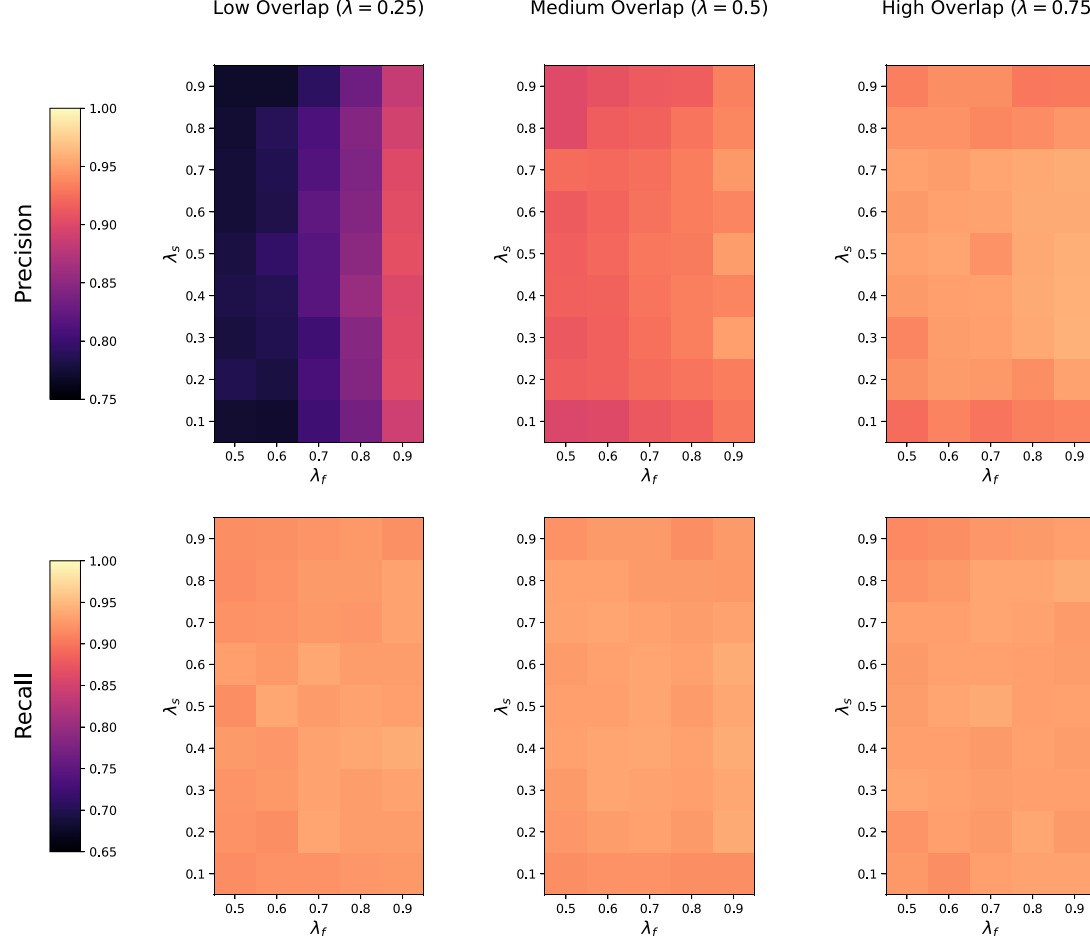

**Appendix 3—figure 2.** Sensitivity of M2S to feature overlap fraction $\lambda$, feature imbalance fraction $\lambda_f$, and sample imbalance fraction $\lambda_s$ between two datasets being matched.

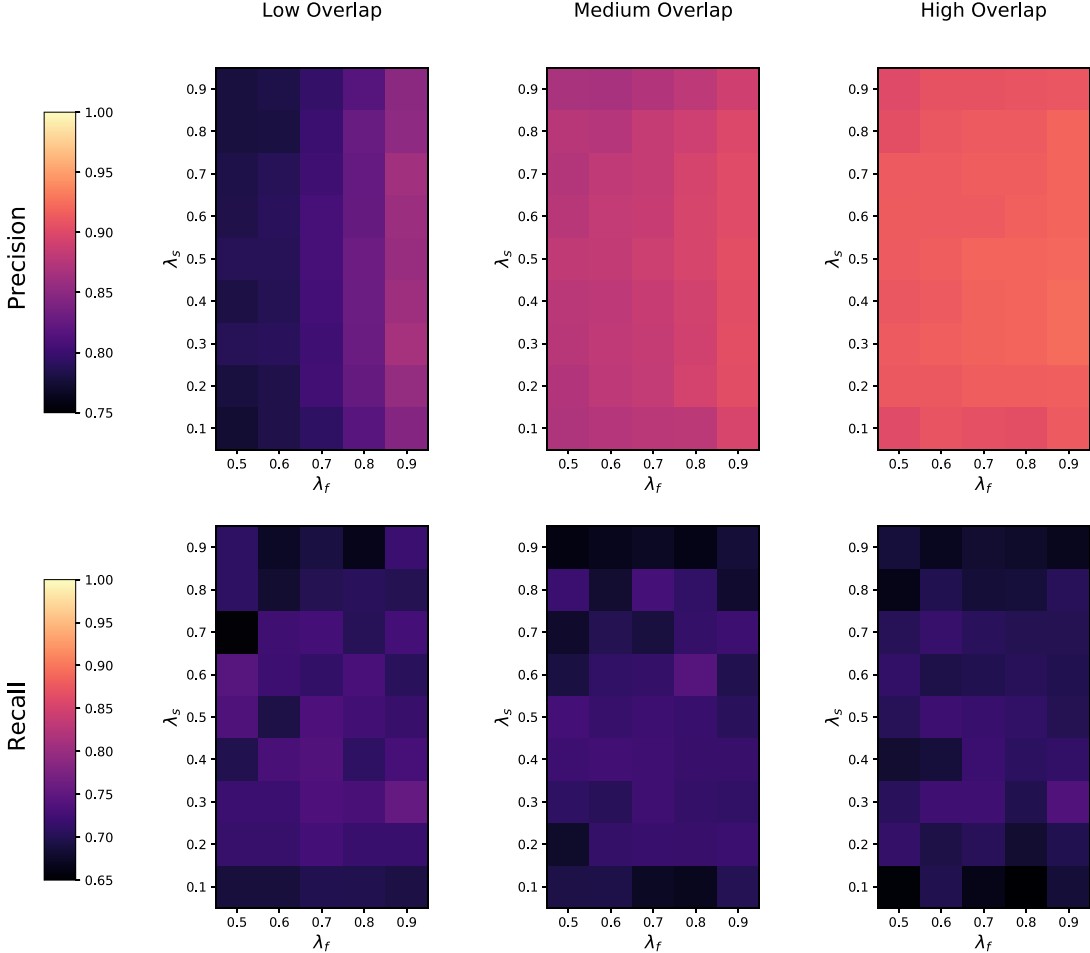

**Appendix 3—figure 3.** Sensitivity of metabCombiner (mC) to feature overlap fraction $\lambda$, feature imbalance fraction $\lambda_f$, and sample imbalance fraction $\lambda_s$ between two datasets being matched.

# Appendix 4

Here we describe additional preprocessing details and analyses of the EPIC data.

## Centered and scaled data - Negative mode

In this section, we present the results obtained on centered and scaled EPIC data in negative mode, shown in *Figure 4* of our main paper. However, due to the smaller size of the validation subset (42 features examined in negative mode compared to 163 in positive mode), the evaluation of the performance of the three methods may be less reliable than in positive mode.

First, we align the CS and HCC studies in negative mode and detect a total of 449, 492, and 180 matches with GM, M2S, and metabCombiner, respectively. Similar to the positive mode analysis, we evaluate the precision and recall of the three methods on the 42 feature validation subset, of which 19 were manually matched. GM and M2S demonstrate identical F1-scores of 0.98, while metabCombiner performs poorly in comparison. GM is able to recover all 19 true matches and identified only 1 false positive, while M2S recovers no false positives but missed 1 true positive.

Next, we align the CS and PC studies in negative mode and detect a total of 485, 569, and 314 matches with GM, M2S, and metabCombiner, respectively. Again, we evaluate the precision and recall of the three methods on the 42 feature validation subset, of which 26 were manually matched. MetabCombiner performs better than in the other EPIC pairings with an F1-score of 0.857, but is still outperformed by the other two methods. GM is slightly outperformed by M2S in this setting, with an almost identical precision of 0.93, but a slightly higher recall for M2S due to detecting 1 additional true positive. However, this remains a good performance for GM since M2S was optimally tuned using the validation subset itself.

## Non-centered and non-scaled data

As a sensitivity analysis, we apply the three methods to EPIC data that has not been centered or scaled. The detailed results can be found in *Appendix 4—table 1*.

**Appendix 4—table 1.** Precision and recall on the EPIC validation subset for unnormalized data in (a) positive mode, and (b) negative mode.
95% confidence intervals were computed using modified Wilson score intervals *Brown et al., 2001*; *Agresti and Coull, 1998*.

| | CS ⟷ HCC | | CS ⟷ PC | |
|---|---|---|---|---|
| Method | Precision | Recall | Precision | Recall |
| GromovMatcher | 0.988 (0.937, 0.999) | 0.944 (0.876, 0.997) | 0.873 (0.776, 0.932) | 0.939 (0.854, 0.976) |
| M2S | 0.967 (0.908, 0.991) | 0.978 (0.923, 0.996) | 0.855 (0.759, 0.917) | 0.985 (0.919, 0.999) |
| metabCombiner | 0.979 (0.889, 0.999) | 0.511 (0.410, 0.612) | 0.926 (0.766, 0.987) | 0.379 (0.271, 0.499) |

(a) Positive mode

| | CS ⟷ HCC | | CS ⟷ PC | |
|---|---|---|---|---|
| Method | Precision | Recall | Precision | Recall |
| GromovMatcher | 0.950 (0.764, 0.997) | 1.000 (0.832, 1.000) | 0.964 (0.823, 0.998) | 0.964 (0.823, 0.998) |
| M2S | 1.000 (0.824, 1.000) | 0.947 (0.754, 0.997) | 0.931 (0.780, 0.988) | 0.964 (0.823, 0.998) |
| metabCombiner | 1.000 (0.566, 1.000) | 0.263 (0.118, 0.488) | 1.000 (0.785, 1.000) | 0.500 (0.326, 0.674) |

(b) Negative mode

M2S was tuned manually on the validation subset to ignore feature intensities in both cases. As a result, it maintains its performance compared to our main experiment. On the other hand, the performance of GM and metabCombiner is affected by the lack of consistency in feature intensities. MetabCombiner's recall drops slightly but its precision remains comparable to that of our main experiment, with the method clearly favoring the latter. Although GM's recall decreases slightly in positive mode, it remains more precise than the optimally tuned M2S, and it balances precision and recall better than metabCombiner. Interestingly, GM's results in negative mode are improved compared to our main experiment, and it outperforms both mC and M2S. However, since

the validation subset in negative mode is relatively small, these differences may not be significant. Nonetheless, GM maintains a good performance, similar to that of the optimally tuned M2S.

Similar to the analysis we conducted on centered and scaled data, we find a high number of false positives when aligning the CS study and the PC study in positive mode. Therefore, we manually examine the matches recovered by GM. Our examination reveals 2 false positives, 4 unclear matches, and 3 additional good matches that GM also identifies in our main analysis. This demonstrates that the lack of centering and scaling results in two additional false positives for GM that are not present in our main results.

## Illustration for alcohol biomarker discovery

*Loftfield et al., 2021* identified 205 features associated with alcohol intake in the CS study, using a false discovery rate (FDR) correction to account for multiple testing. By applying an FDR correction in our pooled analysis, we identify 243 features associated with alcohol intake. Out of those 243 features, 185 are consistent with the features identified in the discovery step of *Loftfield et al., 2021*, while 55 features are newly discovered (*Figure 5c*). We examine the 20 features identified as significant in Loftfield et al.'s discovery analysis but that are not significant in our pooled analysis. Both manual and GM matching yield identical results for these features, indicating that the loss of significance is not due to incorrect matching. Upon further investigation, we find that these features do not demonstrate a meaningful association with alcohol intake in the HCC and PC studies. This observation is reinforced by the fact that none of these features are among the 10 features that persisted after the validation step in Loftfield et al.

Out of the 205 features initially discovered in *Loftfield et al., 2021*, 10 are replicated in the EPIC HCC and PC studies using the more stringent Bonferroni correction. When using a Bonferroni correction in our pooled analysis, we find significant association between alcohol intake and 92 features, 36 of which are effectively shared by the three studies. Notably, these features include all 10 features that were retained in Loftfield et al. (*Figure 5c*).

This analysis illustrates how GromovMatcher can be used in the context of biomarker discovery, and its potential to allow for increased statistical power.

# Appendix 5

Here we investigate how the choice of the reference dataset influences the discovery of metabolites shared across the CS, HCC and PC EPIC studies by GromovMatcher. All three methods considered in this paper, GromovMatcher, M2S, and metabCombiner, are limited to the comparison of two datasets. However, they can still be used to compare and pool multiple datasets using a multi-step procedure. Namely, this can be done by designating a 'reference' dataset and aligning all studies to it one by one. We take this exact approach in our analysis when aligning the CS, HCC, and PC studies of the EPIC data in positive mode. Namely, the HCC and PC studies are both aligned to the CS study (see main text *Figure 5b*). However, this method raises two critical questions: (*i*) how does the use of a reference dataset affect matching results, and (*ii*) how is the matching affected by the choice of reference dataset.

To address these questions, we compare the features identified as common to the three studies using two different studies as references: the CS study used as reference in the main analysis, and the HCC study. For simplicity, let's denote $M_{\text{study 1,study 2}}$ the matching matrix obtained when aligning study 1 and study 2.

## Changes in matching results when reference dataset is used

Concerning question (i), we compare two matchings: HCC to CS to PC (the matrix product $M_{\text{CS,HCC}}^T M_{\text{CS,PC}}$) which we will refer to as the reference matching, and the direct matching of PC to HCC ($M_{\text{HCC,PC}}$). Note that these matchings are not fully comparable as the former considers only features found in CS, potentially missing unique HCC and PC matches. We can however compare the two matchings on the subset of 706 features common to all three studies, as determined by the reference matching. We find that the direct matching supports 683 out of them, indicating that the matching via a reference still yields good results compared to the direct matching (see *Appendix 5—figure 1*).

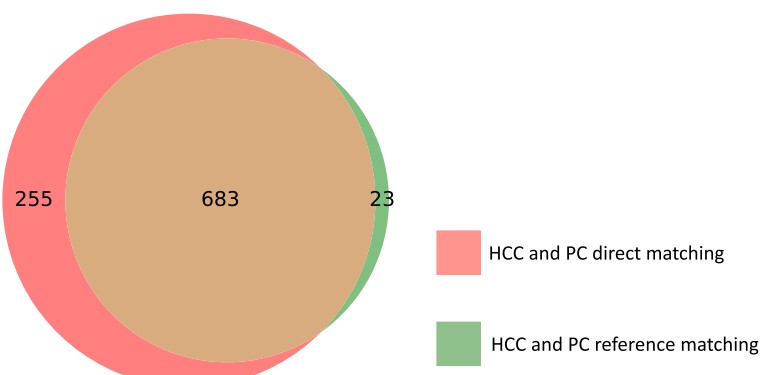

**Appendix 5—figure 1.** Overlap between the 706 features common to the HCC and PC studies found via reference matching, and the 938 features common to HCC and PC found by direct matching.

## Effect of reference dataset choice on matching results

Concerning question (*ii*), we compare the features identified as common to the three studies using two different studies as references: the CS study used in the paper, and the HCC study. We find that they identify 706 and 708 common features respectively, with an overlap of 640 features (see *Appendix 5—figure 2*). This highlights that the choice of reference does matter to some extent. In the paper, choosing CS as a reference was informed by CS's sample size, and study population.

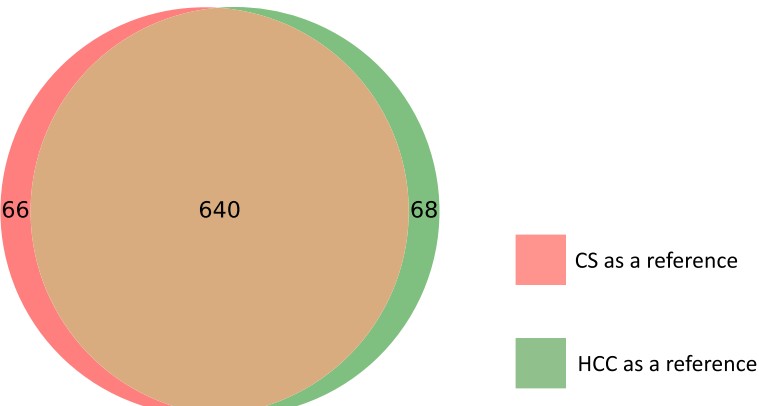

**Appendix 5—figure 2.** Overlap between the features identified as common to the three EPIC studies using either the CS study or the HCC study as a reference.

