## [Editor Report · eLife assessment]

The authors describe an **important** tool, GromovMatcher, that can be used to compare proteomic data from various experimental approaches. The underlying method is innovative, the algorithm is clearly described, and the validation that is presented is **convincing**.

---

## [Referee Report · Reviewer #1 (Public Review)]

Summary:

The authors have implemented Optimal Transport algorithm in GromovMatcher for comparing LC/MS features from different datasets. This paper gains significance in the proteomics field for performing meta-analysis of LC/MS data.

Strengths:

The main strength is that GromovMatcher acheives significant performance metrics compared to other existing methods. The authors have done extensive comparisons to claim that GromovMatcher performs well.

Weaknesses:

The authors might need to add the limitation of datasets and thus have tested/validated their tool using simulated data in the abstract as well.

---

## [Referee Report · Reviewer #2 (Public Review)]

Summary

The goal of untargeted metabolomics is to identify differences between metabolomes of different biological samples.

Untargeted metabolomics identifies features with specific mass-to-charge-ratio (m/z) and retention time (RT). Matching those to specific metabolites based on the model compounds from databases is laborious and not always possible, which is why methods for comparing samples on the level of unmatched features are crucial.

The main purpose of the GromovMatcher method presented here is to merge and compare untargeted metabolomes from different experiments. These larger datasets could then be used to advance biological analyses, for example, for identification of metabolic disease markers.

The main problem that complicates merging different experiments is that m/z and RT vary slightly for the same feature (metabolite).

The main idea behind the GromovMatcher is built on the assumption that if two features match between two datasets (that feature i from dataset 1 matches feature j from dataset 2, and feature k from dataset 1 matches feature l from dataset 2), then the correlations or distances between the two features within each of the datasets (i and k, and j and l) will be similar. The authors then use the Gromov-Wasserstein method to find the best matches matrix from these data.

The variation in m/z between the same features in different experiments is a user-defined value and it is initially set to 0.01 ppm. There is no clear limit for RT deviations, so the method estimates a non-linear deviation (drift) of RT between two studies. GromovMatcher estimates the drift between two studies, and then discards the matching pairs where the drift would deviate significantly from the estimate. It learns the drift from a weighted spline regression.

The authors validate the performance of their GromovMatcher method using a dataset of cord blood. They use 20 different splits and compare the GromovMatcher (both its GM and GMT iterations, whereby GMT version uses the deviation from estimated RT drift to filter the matching matrix) with two other matching methods: M2S and metabCombiner.

The second validation was done using a (scaled and centered) dataset of metabolics from cancer datasets from the EPIC cohort that were manually matched by an expert. This dataset was also used to show that using automated methods can identify more features that are associated with a particular group of samples than what was found by manual matching. Specifically, the authors identify additional features connected to alcohol consumption.

Strengths:

I see the main strength of this work in its combination of all levels of information (m/z, RT, and higher-order information on correlations between features) and using each of the types of information in a way that is appropriate for the measure. The most innovative aspect is using the Gromov-Wasserstein method to match the features based on distance matrices.

The authors of the paper identify two main shortcomings with previously established methods that attempt to match features from different experiments: (a) all other methods require fine-tuning of user-defined parameters, and, more importantly, (b) do not consider correlations between features. The main strength of the GromovMatcher is that it incorporates the information on distances between the features (in addition to also using m/z and RT).

Weaknesses:

The main weakness is that there seem not to be enough manually curated datasets that could be used for validation. It will, therefore, be important, for the authors, and the field in general to keep validating and improving their methods if more datasets become available.

The second weakness, as emphasized by the authors in the discussion is that the method as it is set up now can be directly used only to compare two datasets. I am confident that the authors will successfully implement novel algorithms to address this issue in the future.

---

## [Author Response]

The following is the authors’ response to the original reviews.

eLife assessmentThis paper represents important findings when identifying untargeted metabolomics and its differences between metabolomes of different biological samples. GromovMatcher is the fantasy name for the soft development. The main idea behind it is built on the assumption of featuring and matching complex datasets. Although the manuscript reflects a solid analysis, it remains incomplete for validation with putative non-curated datasets.

We are grateful to the eLife editor for taking the time and effort to assess our manuscript.

We are however unsure of what the editor means by “it remains incomplete for validation with putative non-curated datasets”. As noted by Reviewer 2, manually curated datasets that could be used for validation are scarce. Most publicly available datasets do not contain sufficient information to establish a ground truth matching on which GromovMatcher, M2S, or metabCombiner can be tested. Even in the case where such a ground truth matching can be established, it must be performed by-hand through a manual matching process which is extremely time-consuming and requires very specific expertise. This, in our opinion, only highlights the need for automatic alignment methods such as metabCombiner, M2S or GromovMatcher.

We do agree that the performance of GromovMatcher (and its competitors) needs to be validated further, and we plan to continue validating GromovMatcher as additional data becomes available in EPIC and other cohorts. With that in mind, the lack of publicly available validation data is the reason why we conducted such an extensive simulation study, arguably more comprehensive than previous validations, exploring challenging settings that we believe reflect real-life scenarios (main text “Validation on ground-truth data” and Appendix 3). We would like to stress that this allows us to highlight previously ignored limitations of the previously published methods, metabCombiner and M2S.

We wish to thank the editor and reviewers for their time and efforts in reviewing our manuscript which led to many significant additions to our paper. Namely we:

• Performed an additional sensitivity analysis (Appendix 3) exploring how an imbalance in the number of features or samples between two studies being matched (e.g. the dataset split), affects the quality of matchings found by GromovMatcher, metabCombiner, and M2S.

• Investigated how changing or removing the reference dataset (Appendix 5) in the EPIC study (main text “Application to EPIC data”), affects the results of GromovMatcher.

• Improved alignment matrix visualizations in Fig. 3a for all four methods tested on the validation data, to highlight more clearly which feature matches were correctly identified or missed.

The revised paper is uploaded as the file “main_elife_revision.pdf” where all revisions are highlighted in blue as well as a copy “main_elife_revision_nohighlights.pdf” where revisions are not highlighted.

**Public Reviews:**

**Reviewer #1 (Public Review):**

Summary:

The authors have implemented the Optimal Transport algorithm in GromovMatcher for comparing LC/MS features from different datasets. This paper gains significance in the proteomics field for performing meta-analysis of LC/MS data.Strengths:The main strength is that GromovMatcher achieves significant performance metrics compared to other existing methods. The authors have done extensive comparisons to claim that GromovMatcher performs well.Weaknesses:There are two weaknesses.(1) When the number of features is reduced the precision drops to ~0.8.

We would like to clarify that this drop in precision occurs in the challenging setting where only a small proportion of metabolites are shared between both datasets (e.g., the overlap – or proportion of shared features - was 25% in our simulation study). When two untargeted metabolic datasets share only 25% of their features, this is a challenging setting for any automated matching method as the vast majority 75% of the features in both datasets must remain unmatched.

In such settings, the reviewer correctly observes that the precision of GromovMatcher algorithms (GM and GMT) drops within the range of 0.80 - 0.85 (Figure 3b, top left panel). Such a precision of 0.8 or larger is still competitive compared with the alternative methods MetabCombiner (mC) and M2S whose precisions drop below 0.8 (see main text Fig. 3b, top left panel).

Precision is measured as the number of metabolite pairs correctly matched divided by all matches identified by a method. In other words, even in the challenging setting when the number of shared features (true matches) between both datasets is small (e.g. low 25% overlap), upwards of 80% of the feature matches found by GromovMatcher are correct which is a very encouraging result.

(2) How applicable is the method for other non-human datasets?

We thank the reviewer for raising this question. The crux of the matter concerning the application to animal data revolves around the hypothesis that correlations between metabolites in two different studies are preserved. Theoretically, the metabolome operates under similar principles in humans, governed by an underlying network of biochemical reactions. Consequently, in comparable human populations, the GM hypothesis is likely to hold to some extent.

However, in practice, application to animal data is more complicated. Animal studies tend to have smaller sample sizes and often stem from intervention-driven scenarios, such as mice subjected to specific diets or chemicals. This results in deliberate alterations in metabolic structures which makes finding two comparable animal studies less likely. To investigate the reviewer’s question, we have searched through the two predominant LC-MS dataset repositories (MetaboLights and NIH Metabolomics Workbench) but did not find any pairs of comparable animal studies due to the reasons mentioned above. One potential strategy to navigate this issue could entail regressing the metabolic intensities against the variables that notably differ between the two animal populations and running GM using the residual intensities. This would be an interesting direction for future research and additional validation would be needed to test the robustness of GM in this setting.

**Reviewer #2 (Public Review):**
Summary:The goal of untargeted metabolomics is to identify differences between metabolomes of different biological samples. Untargeted metabolomics identifies features with specific mass-to-charge ratio (m/z) and retention time (RT). Matching those to specific metabolites based on the model compounds from databases is laborious and not always possible, which is why methods for comparing samples on the level of unmatched features are crucial.The main purpose of the GromovMatcher method presented here is to merge and compare untargeted metabolomes from different experiments. These larger datasets could then be used to advance biological analyses, for example, for the identification of metabolic disease markers. The main problem that complicates merging different experiments is m/z and RT vary slightly for the same feature (metabolite).The main idea behind the GromovMatcher is built on the assumption that if two features match between two datasets (that feature I from dataset 1 matches feature j from dataset 2, and feature k from dataset 1 matches feature l from dataset 2), then the correlations or distances between the two features within each of the datasets (i and k, and j and l) will be similar. The authors then use the Gromov-Wasserstein method to find the best matches matrix from these data.The variation in m/z between the same features in different experiments is a user-defined value and it is initially set to 0.01 ppm. There is no clear limit for RT deviations, so the method estimates a non-linear deviation (drift) of RT between two studies. GromovMatcher estimates the drift between the two studies and then discards the matching pairs where the drift would deviate significantly from the estimate. It learns the drift from a weighted spline regression.The authors validate the’performance of their GromovMatcher method by a validation experiment using a dataset of cord blood. They use 20 different splits and compare the GromovMatcher (both its GM and GMT iterations, whereby the GMT version uses the deviation from estimated RT drift to filter the matching matrix) with two other matching methods: M2S and metabCombiner.The second validation was done using a (scaled and centered) dataset of metabolics from cancer datasets from the EPIC cohort that was manually matched by an expert. This dataset was also used to show that using automatic methods can identify more features that are associated with a particular group of samples than what was found by manual matching. Specifically, the authors identify additional features connected to alcohol consumption.Strengths:I see the main strength of this work in its combination of all levels of information (m/z, RT, and higher-order information on correlations between features) and using each of the types of information in a way that is appropriate for the measure. The most innovative aspect is using the Gromov-Wasserstein method to match the features based on distance matrices.

We thank the reviewer for acknowledging this strength of our proposed GromovMatcher method.

The authors of the paper identify two main shortcomings with previously established methods that attempt to match features from different experiments: (a) all other methods require fine-tuning of user-defined parameters, and, more importantly, (b) do not consider correlations between features. The main strength of the GromovMatcher is that it incorporates the information on distances between the features (in addition to also using m/z and RT).Weaknesses:The first, minor, weakness I could identify is that there seem not to be plenty of manually curated datasets that could be used for validation.

We thank the reviewer for raising this issue concerning manually curated validation data.

Manually curated datasets available for validation purposes are indeed scarce. This stems from the laborious nature of matching features across diverse studies, hence the need for automatic matching methods. Our future strategy involves further validation of the GromovMatcher approach as more data becomes accessible in EPIC and other cohorts.

The scarcity of real-life publicly available datasets that can be used for validation purpose is the reason why we conducted an extensive simulation study (main text “Validation on ground-truth data” and Appendix 3). It is notably thorough, arguably more comprehensive than previous validations, utilizes real-life untargeted data, and imitates situations where data originates from distinct untargeted metabolomics studies, complete with realistic noise parameters encompassing RT, mz, and feature intensities. Our validation study comprehensively explores the performance of GromovMatcher, M2S, and metabCombiner, including in challenging realistic settings where there is a nonlinear drift in retention times, varying levels of feature overlaps between studies, normalizations of feature intensities, as well as imbalances in the number of features and samples present in the studies being matched.

The second is also emphasized by the authors in the discussion. Namely, the method as it is set up now can be directly used only to compare two datasets.

This is indeed a limitation that is common to all three methods considered in this paper. However, all these methods, GromovMatcher, M2S, and metabCombiner, can still be used to compare and pool multiple datasets using a multi-step procedure. Namely, this can be done by designating a 'reference' dataset and aligning all studies to it one by one. We take this exact approach in our paper when aligning the CS, HCC, and PC studies of the EPIC data in positive mode (main text “Application to EPIC data”). Namely, the HCC and PC studies are both aligned to the CS study by running GromovMatcher twice, and after obtaining these matchings, our analysis is restricted to those features in HCC and PC that are present in the CS study.

After the reviewer’s comment, we have added an additional sensitivity analysis in Appendix 5, to compare the results produced by GromovMatcher depending on the choice of the reference study. Namely, setting the reference study to either the CS study or the HCC study, GromovMatcher identified 706 and 708 common features respectively, with an overlap of 640 features. This highlights that the choice of reference does matter to some extent. In our original analysis of the EPIC data, choosing CS as the reference was motivated by the fact that CS had the largest sample size (compared to HCC and PC) and a subset of features in HCC and PC were already matched by experts to the CS study which we could use for validation (see Loftfield et al. (2021). J Natl Cancer Inst.).

As mentioned in the discussion section of our manuscript, the recently proposed multimarginal Gromov-Wasserstein algorithm (Beier, F., Beinert, R., & Steidl, G. (2023). Information and Inference) could potentially allow multiple metabolomic studies to be matched using one optimization routine (e.g. without the designation of a ‘reference study’ for matching). We have not explored this possibility in depth yet as fast numerical methods for multimarginal GW are still in their infancy. Also, such multimarginal methods rely on the computation and storage of coupling or matching matrices that are tensors where the number of dimensions is equal to the number of datasets being matched. Therefore, multimarginal methods have large memory costs, which currently precludes their application for the matching of multiple metabolomics datasets.

**Reviewer #2 (Recommendations For The Authors):**
(1) I was struggling with the representation used in Figure 3a. The gray points overlayed over the green points on a straight line are difficult to visually quantify. I found that my eyes mainly focused on the pattern of the red dots.

Figure 3a has been modified to improve visual clarity. Namely we have consistently reordered the rows and columns of the coupling matrices such that the true positive matches (green points) are spatially separated from the false negative matches (red points). Now the fraction of true positive and false negative matches can be appreciated much more clearly by eye in Figure 3a.

(2) I would also like to add the caveat that I cannot judge whether the authors used the other two methods that they compare with GromovMatcher (the M2S and metabCombiner) optimally. But I also do not see any evidence that they did not. Hopefully one of the other reviewers can address that.

We appreciate the reviewer for highlighting the comparison of our approach GromovMatcher to the other existing methods M2S and MetabCombiner (mC). Both M2S and mC depend on tens of hyperparameters each with a discrete or continuous set of values that must be properly optimized to infer accurate matchings between dataset features. We detail in Appendix 2 how the hyperparameters of the M2S and mC methods are optimally tuned to achieve the best possible performance on the validation ground-truth data. Namely, both in the simulation study and on EPIC data, we grid-search over all important hyperparameters in the M2S and mC methods and choose those parameter combinations that result in the highest F1 score, averaged over 20 random trials. We remark that no such hyperparameter optimization was performed for our GromovMatcher method. As shown in Figures 3 and 4 of the main text, we find that GromovMatcher outperforms M2S and mC even in these cases when the hyperparameters of M2S and mC are tuned to predict optimal feature matchings.

Given the large combinatorial space of hyperparameter choices, we believe we have thoroughly tested the important hyperparameter combinations that users of M2S and mC would be likely to explore in their own research.

(3) Validation(3a) The first validation is done on a split cord blood dataset. I could not clearly see from the paper how sensitive the result is to the dataset split.

We are grateful for the reviewer’s question and have included new experiments in Appendix 3 which show how the results of GromovMatcher, M2S, and MetabCombiner are affected by the dataset split. In our original manuscript, our validation ground-truth experiment began with an untargeted metabolomic dataset consisting of n = 499 samples and p = 4,712 metabolic features which is split equally into two datasets consisting of an equal number of samples n1 = n2 and an equal number of metabolic features p1 = p2. The features of these equal-sized datasets would then be matched by our method.

Now in Appendix 3 (Figs. 1-3) we show the sensitivity of all three alignment methods (GromovMatcher, M2S, and MetabCombiner) when we vary the fraction of samples in dataset 1 over dataset 2 given by n1/ n2, the overlap in shared features between both datasets, and the fraction of metabolic features in dataset 1 that are not present in dataset 2 which affects the feature sizes of both datasets p1/ p2. We find that all alignment methods are able to maintain a consistent precision and recall score when these three dataset split parameters are varied. GromovMatcher achieves a higher precision and recall than M2S and MetabCombiner for all choices of dataset split, agreeing with the validation experiment results from the main text (see main text Fig. 3). All three methods tested decrease in precision (without dropping in recall) when dataset 1 and dataset 2 contain an equal number of unshared features (e.g. when p1 = p2). Therefore, these sensitivity experiments in Appendix 3 show that our results in the main text are performed in the most challenging setting for the dataset split.

(3b) The second validation was done using a (scaled and centered) dataset of metabolics from cancer datasets from the EPIC cohort that was manually matched by an expert. Here the authors observe that metabCombiner has good precision, but lags in recall. And M2S has a very similar performance to GromovMatcher. The authors explain this by the fact that the drift in RT between the two experiments is mostly linear and thus does not affect the M2S performance. Can the authors find a different validation dataset where the drift in RT is not linear? If yes, it would be interesting to add it to the paper.

We thank the reviewer for raising this question. As mentioned above, curated validation datasets such as the EPIC study analyzed in our paper are very rare and we do not currently have a validation study with a nonlinear retention time drift.

Nevertheless, we performed an additional analysis of simulated data (reported in Appendix 2 – “M2S hyperparameter experiments” and Appendix 2 – Table 1) that demonstrates the decrease in M2S performance when the simulated drift is nonlinear. As presented in Appendix 2 – Table 1, in a low overlap setting with a linear drift which corresponds to the EPIC data, precision and recall were 0.831 and 0.934 respectively, instead of 0.769 and 0.905 in the main analysis where the drift was nonlinear.